# Receding Horizon Inverse Reinforcement Learning

**Yiqing Xu**[1] **Wei Gao**[1] **David Hsu**[1,2]

[1]School of Computing
[2]Smart Systems Insitute
National University of Singapore
{xuyiqing,gaowei90,dyhsu}@comp.nus.edu.sg

## Abstract

Inverse reinforcement learning (IRL) seeks to infer a cost function that explains the underlying goals and preferences of expert demonstrations. This paper presents *receding-horizon inverse reinforcement learning* (RHIRL), an IRL algorithm for high-dimensional, noisy, continuous systems with black-box dynamic models. RHIRL addresses two key challenges of IRL: scalability and robustness. To handle high-dimensional continuous systems, RHIRL matches the induced optimal trajectories with expert demonstrations *locally* in a receding horizon manner and "stitches" together the local solutions to learn the cost; it thereby avoids the "curse of dimensionality". This contrasts with earlier algorithms that match with expert demonstrations *globally* over the entire high-dimensional state space. To be robust against imperfect expert demonstrations and control noise, RHIRL learns a state-dependent cost function "disentangled" from system dynamics under mild conditions. Experiments on benchmark tasks show that RHIRL outperforms several leading IRL algorithms in most instances. We also prove that the cumulative error of RHIRL grows linearly with the task duration.

## 1 Introduction

Reinforcement learning (RL) has made exciting progress in a range of complex tasks, including real-time game playing [24], visuo-motor control of robots [35], and many other works. The success, however, often hinges on a carefully crafted cost function [17, 25], which is a major impediment to the wide adoption of RL in practice. Inverse reinforcement learning (IRL) [26] addresses this need by learning a cost function that explains the underlying goals and preferences of expert demonstrations. This work focuses on two key challenges in IRL, *scalability* and *robustness*.

Classic IRL algorithms commonly consist of two nested loops. The inner loop approximates the optimal control policy for a hypothesized cost function, while the outer loop updates the cost function by comparing the behavior of the induced policy with expert demonstrations. The inner loop must solve the (forward) reinforcement learning or optimal control problem, which is in itself a challenge for complex high-dimensional systems. Many interesting ideas have been proposed for IRL, including, e.g., maximum entropy learning [39, 41], guided cost learning [7], and adversarial learning [8]. See Figure 1 for illustrations. They try to match a *globally* optimal approximate policy with expert demonstrations over the entire system state space or a sampled approximation of it. This is impractical for high-dimensional continuous systems and is a fundamental impediment to scalability. Like RL, IRL suffers from the same "curse of dimensionality". To scale up, *receding-horizon IRL* (RHIRL) computes locally optimal policies with receding horizons rather than a globally optimal policy and then matches them with expert demonstrations *locally* in succession (Figure 1*d*). The local approximation and matching substantially mitigate the impact of high-dimensional space and improve the sample efficiency of RHIRL, at the cost of a local rather than a global solution. So RHIRL trades off optimality for scalability and provides an alternative to current approaches.

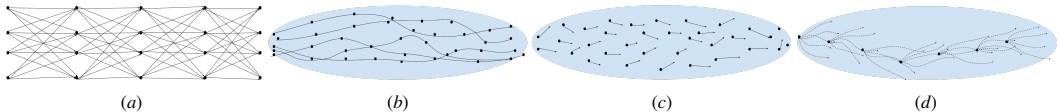

(a)         (b)         (c)         (d)

Figure 1: A comparison of RHIRL and selected IRL algorithms. They all try to match the policy induced by the learned cost with expert demonstrations. (*a*) MaxEnt matches the exact feature count over the entire system state space. (*b*) REIRL, GCL, and GAN-GCL match the approximate feature count over sampled state trajectories globally over the entire task duration. (*c*) GAIL, AIRL, ... discriminate the sampled state or state-action distributions. (*d*) RHIRL matches control sequences *locally* along demonstrated trajectories in a receding horizon manner.

Another important concern of IRL is noise in expert demonstrations and system control. Human experts may be imperfect for various reasons and provide good, but still suboptimal demonstrations. Further, the system may fail to execute the commanded actions accurately because of control noise. We want to learn a cost function that captures the expert's intended actions rather than the imperfectly executed actions. While earlier work has investigated the question of learning from sub-optimal or failed demonstrations [3, 31, 38], there is a subtle, but critical difference between (i) demonstrations intended to be suboptimal and (ii) optimal demonstrations corrupted by noise. The existing work [3, 31, 38] addresses (i); RHIRL addresses (ii). To learn the true intentions from the noise corrupted demonstrations, RHIRL relies on a simplifying assumption: the cost function is linearly separable with two components, one state-dependent and one control-dependent. Many interesting systems in practice satisfy the assumption, at least, approximately [28]. RHIRL then learns the state-dependent cost component, which is disentangled from the system dynamics [8] and agnostic to noise.

## 2 Related Work

IRL can be viewed as an indirect approach to imitation learning. It learns a cost function, which induces an optimal policy whose behavior matches with expert demonstrations. In contrast, behavior cloning (BC) is a direct approach. It assumes independence among all demonstrated state-action pairs and learns a policy that maps states to actions through supervised learning of state-action pairs from expert demonstrations. The simplicity of BC is appealing. However, it typically requires large amounts of data to learn well and suffers from covariate shift [30]. IRL is more data-efficient. Further, it produces a cost function, which explains the expert demonstrations and potentially transfers to other systems with different dynamics. These benefits, however, come at the expense of greater computational complexity.

Classic IRL algorithms learn a cost function iteratively in a double-loop: the outer loop updates a hypothesized cost function, and the inner loop solves the forward RL problem for an optimal policy and matches it with expert demonstrations. Various methods have been proposed [1, 7, 39, 41], but they all seek a globally optimal solution over the entire state space (Figure 1*a*) [39, 41] or the entire task duration (Figure 1*b*) [1, 7]. As a result, they face significant challenges in scalability and must make simplifications, such as locally linear dynamics [7]. Recent methods use the generative adversarial network (GAN) [10] to learn a discriminator that differentiates between the state or state-action distribution induced by the learned cost and that from expert demonstrations [5, 8, 9, 13, 14, 18, 21, 27, 29]. We view this global matching as a major obstacle to scalability. In addition, GAN training is challenging and faces difficulties with convergence.

RHIRL stands in between BC and the aforementioned IRL algorithms by trading off optimality for scalability. BC performs local matching at each demonstrated state-action pair, treating all of them independently. Most existing IRL algorithms perform global matching over the entire state space or sampled trajectories from it. RHIRL follows the standard IRL setup. To tackle the challenge of scalability for high-dimensional continuous systems, RHIRL borrows ideas from *receding horizon control* (RHC) [19]. It solves for locally optimal control sequences with receding horizons and learns the cost function by "stitching" together a series of locally optimal solutions to match the global state distribution of expert demonstrations (Figure 1*d*). Earlier work has explored the idea of RHC in IRL [23], but relies on handcrafted reward features and is limited to discrete, low-dimensional tasks. More recent work learns reward features automatically [20]. However, it focuses on lane navigation for autonomous driving, exploiting a known analytic model of dynamics and noise-free,

perfect expert demonstrations. RHIRL aims at general high-dimensional continuous tasks with noisey expert demonstrations.

Another challenge to IRL is suboptimal expert demonstrations and system control noise. Several methods learn an auxiliary score or ranking to reweigh the demonstrations, in order to approximate the underlying optimal expert distribution [4, 5, 38]. RHIRL does not attempt to reconstruct the optimal expert demonstrations. It explicitly models the human errors in control actions as additive Gaussian noise and matches the noisy control with expert demonstrations, in order to learn from the intended, rather than the executed expert actions. Modelling the human error as additive Gaussian noise is a natural choice technically [22, 34, 40], but compromises on realism. Human errors in sequential decision making may admit other structural regularities, as a result of planning, resource constraints, uncertainty, or bounded rationality[15]. Studying the specific forms of human errors in the IRL context requires insights beyond the scope of our current work, but forms a promising direction for future investigation.

## 3   Receding Horizon Inverse Reinforcement Learning

### 3.1   Overview

Consider a continuous dynamical system:

$$x_{t+1} = f(x_t, v_t), \tag{1}$$

where $x_t \in \mathcal{R}^n$ is the state, $v_t \in \mathcal{R}^m$ is the control at time $t$, and the initial system state at $t = 0$ follows a distribution $\mu$. To account for noise in expert demonstrations, we assume that $v_t$ is a random variable following the Gaussian distribution $\mathcal{N}(v_t|u_t, \Sigma)$, with mean $u_t$ and covariance $\Sigma$. We can control $u_t$ directly, but not $v_t$, because of noise. The state-transition function $f$ captures the system dynamics. RHIRL represents $f$ as a black-box simulator and does not require its analytic form. Thus, we can accommodate arbitrary complex nonlinear dynamics. To simplify the presentation, we assume that the system dynamics is deterministic. We sketch the extension to stochastic dynamics at the end of the section, the full proof is given in Appendix C.

In RL, we are given a cost function and want to find a control policy that minimizes the expected total cost over time under the dynamical system. In IRL, we are not given the cost function, but instead, a set $\mathcal{D}$ of expert demonstrations. Each demonstration is a trajectory of states visited by the expert over time: $\tau = (x_0, x_1, x_2, ..., x_{T-1})$ for a duration of $T$ steps.

We assume that the expert chooses the actions to minimize an unknown cost function and want to recover this cost from the demonstrations. Formally, suppose that the cost function is parameterized by $\theta$. RHIRL aims to learn a cost function that minimizes the loss $\mathcal{L}(\theta; \mathcal{D})$, which measures the difference between the demonstration trajectories and the optimal control policy induced by the cost function with parameters $\theta$.

RHIRL performs this minimization iteratively, using the gradient $\partial \mathcal{L}/\partial \theta$ to update $\theta$. In iteration $t$, let $x_t$ be the system state at time $t$ and $\mathcal{D}_t$ be the set of expert sub-trajectories starting at time $t$ and having a duration of maximum $K$ steps. We use the current cost to perform receding horizon control (RHC) at $x_t$, with time horizon $K$, and then update the cost by comparing the resulting state trajectory distribution with the demonstrations in $\mathcal{D}_t$. Earlier IRL work requires the simulator to reset to the states from the initial state distribution [1, 6, 7, 8, 13, 18]. To perform receding horizon optimization, RHIRL requires a simulator to reset to the initial state $x_t$ for the local control sequence optimization at each time step $t = 0, \ldots T - 1$, a common setting in optimal control (e.g., [36, 37, 20]). See Algorithm 1 for a sketch.

First, sample $M$ control sequences, each of length $K$ (line 6). We assume that the covariance $\Sigma$ is known. If it is unknown, we set it to be identity by default. For our experiment results reported in Table 1 and Table 2, $\Sigma$ is unknown and is approximated by a constant factor of the identity matrix (grid search is performed to determine this constant factor). We show through experiments that the learned cost function is robust over different noise settings (Section 4.3). Next, we apply model-predictive path integral (MPPI) control [37] at $x_t$. MPPI provides an analytic solution for the optimal control sequence distribution and the associated state sequence distribution, which allow us to estimate the gradient $\partial \mathcal{L}/\partial \theta$ efficiently through importance sampling (lines 7–8) and update cost function parameters $\theta$ (line 9). Finally, we execute the computed optimal control (line 10–12)

---
**Algorithm 1** RHIRL
---
1: Initialize $\theta$ randomly.
2: **for** $i = 1, 2, 3, \dots$ **do**
3:     Sample $x_0$ from $\mu$.
4:     Initialize control sequence $U$ of length $K$ to $(0, 0, \dots)$.
5:     **for** $t = 0, 1, 2, \dots, T - 1$ **do**
6:         Sample $M$ control sequences $V_j$ for $j = 1, 2, \dots M$, according to $\mathcal{N}(V|U, \Sigma)$.
7:         Compute the importance weight $w_j$ for each $V_j$, using the state cost $S(V_j, x_t; \theta)$. See equation (11).
8:         Compute $\frac{\partial}{\partial \theta}\mathcal{L}(\theta; \mathcal{D}_t, x_t)$ using $\mathcal{D}_t$ and $V_j$ with weight $w_j$, for $j = 1, 2, \dots, M$. See equation (12).
9:         $\theta \leftarrow \theta - \alpha \frac{\partial \mathcal{L}}{\partial \theta}$.
10:        $U \leftarrow \sum_{j=1}^{M} w_j V_j$.
11:        Sample $v_t$ from $\mathcal{N}(v|u, \Sigma)$, where $u$ is the first element in the control sequence $U$.
12:        $x_{t+1} \leftarrow f(x_t, v_t)$.
13:        Remove $u$ from $U$. Append 0 at the end of $U$.
14:     **end for**
15: **end for**
---

and update the mean control input for the next iteration (line 13). We would like to emphasize that we only use the simulator to sample rollouts and evaluate the current cost function. Fixed expert demonstrations $\mathcal{D}$ are given upfront. Unlike DAgger, we do not query the expert online during training.

Our challenge is to uncover a cost function that captures the expert's intended controls $(u_0, u_1, \dots, u_{T-1})$, even though they were not directly observed, because of noise, and do so in a scalable and robust manner for high-dimensional, noisy systems.

We develop three ideas: structuring the cost function, matching locally with expert demonstrations, and efficient computation of the gradient $\partial \mathcal{L}/\partial \theta$, which are described next.

### 3.2 Robust Cost Functions

To learn a cost function robust against noise, we make a simplifying assumption that linearly separates the one-step cost into two components: a state cost $g(x; \theta)$ parameterized by $\theta$ and a quadratic control cost $u^T \Sigma^{-1} u$. Despite the simplification, this cost function models a wide variety of interesting systems in practice [28]. It allows RHIRL to learn a state cost $g(x; \theta)$, independent of control noise (Section 3.4), and thus generalize over different noise distributions (Section 4.3).

Suppose that $V = (v_0, v_1, \dots, v_{K-1})$ is a control sequence of length $K$, conditioned on the input $U = (u_0, u_1, \dots, u_{K-1})$. We apply $V$ to the dynamical system in (1) with start state $x_0$ and obtain a state sequence $\tau = (x_0, x_1, \dots, x_K)$ with $x_k = f(x_{k-1}, v_{k-1})$ for $k = 1, 2, \dots, K$. Define the total cost of $V$ as

$$J(V, x_0; \theta) = \sum_{k=0}^{K} g(x_k; \theta) + \sum_{k=0}^{K-1} \frac{\lambda}{2} u_k^\mathsf{T} \Sigma^{-1} u_k, \tag{2}$$

where $\lambda \geq 0$ is a constant weighting the relative importance between the state and control costs. For convenience, define also the total state cost of $V$ as

$$S(V, x_0; \theta) = \sum_{k=0}^{K} g(x_k; \theta). \tag{3}$$

While $S$ is defined in terms of the control sequence $V$, it only depends on the corresponding state trajectory $\tau$. This is very useful, as the training data contains state and not control sequences explicitly.

### 3.3 Local Control Sequence Matching

To minimize the loss $\mathcal{L}$, each iteration of RHIRL applies RHC with time horizon $K$ under the current cost parameters $\theta$ and computes locally optimal control sequences of length $K$. In contrast, classic IRL algorithms, such as MaxEnt [41], perform global optimization over the entire task duration $T$ in the inner loop. While RHC sacrifices global optimality, it is much more scalable and enables RHIRL

to handle high-dimensional continuous systems. We use the hyperparameter $K$ to trade off optimality and scalability.

Specifically, we use MPPI [37] to solve for an optimal control sequence distribution at the current start state $x_t$ in iteration $t$. The main result of MPPI suggests that the optimal control sequence distribution $Q$ minimizes the "free energy" of the dynamical system and this free energy can be calculated from the cost of the state trajectory under $Q$. Mathematically, the probability density $p_Q(V^*)$ can be expressed as a function of the state cost $S(V, x_t; \theta)$, with respect to a Gaussian "base" distribution $B(U_B, \Sigma)$ that depends on the control cost:

$$p_Q(V^* | U_B, \Sigma, x_t; \theta) = \frac{1}{Z} \, p_B(V^* | U_B, \Sigma) \exp\left(-\frac{1}{\lambda} S(V^*, x_t; \theta)\right), \tag{4}$$

where $Z$ is the partition function. For the quadratic control cost in (2), we have $U_B = (0, 0, \dots)$ [37]. Intuitively, the expression in (4) says that the probability density of $V^*$ is the product of two factors, one penalizing high control cost and one penalizing high state cost. So controls with large values or resulting in high-cost states occur with probability exponentially small.

Equation (4) provides the optimal control sequence distribution under the current cost. Suppose that the control sequences for expert demonstrations $\mathcal{D}_t$ follow a distribution $E$. We define the loss $\mathcal{L}(\theta; \mathcal{D}_t, x_t)$ as the KL-divergence between the two distributions:

$$\mathcal{L}(\theta; \mathcal{D}_t, x_t) = D_{\text{KL}}\big(p_E(V | x_t) \,\|\, p_Q(V | U_B, \Sigma, x_t; \theta)\big), \tag{5}$$

which RHIRL seeks to minimize in each iteration. While the loss $\mathcal{L}$ is defined in terms of control sequence distributions, the expert demonstrations $\mathcal{D}$ provide state information and not control information. However, each control sequence $V$ induces a corresponding state sequence $\tau$ for a given start state $x_0$, and $\tau$ determines the cost of $V$ according to (2). We show in the next subsection that $\partial \mathcal{L}/\partial \theta$ can be computed efficiently using only state information from $\mathcal{D}$. This makes RHIRL appealing for learning tasks in which control labels are difficult or impossible to acquire.

## 3.4 Gradient Optimization

To simplify notations, we remove the explicit dependency on $x_t$ and $\mathcal{D}_t$ and assume that in iteration $t$ of RHIRL, all control sequences are applied with $x_t$ as the start state and expert demonstrations come from $\mathcal{D}_t$. We have

$$\begin{aligned}
\frac{\partial \mathcal{L}}{\partial \theta} &= \frac{\partial}{\partial \theta} \int p_E(V) \log \frac{p_E(V)}{p_Q(V | U_B, \Sigma; \theta)} dV \\
&= \int p_E(V) \left(\frac{1}{\lambda} \frac{\partial}{\partial \theta} S(V; \theta)\right) dV - \int p_Q(V | U_B, \Sigma; \theta) \left(\frac{1}{\lambda} \frac{\partial}{\partial \theta} S(V; \theta)\right) dV
\end{aligned} \tag{6}$$

The first line follows directly from the definition, and the derivation for the second line is available in Appendix A.

We estimate the two integrals in (6) through sampling. For the first integral, we can use the expert demonstrations as samples. For the second integral, we cannot sample $p_Q$ directly, as the optimal control distribution $Q$ is unknown in advance. Instead, we sample from a known Gaussian distribution with density $p(V | U, \Sigma)$ and apply importance sampling so that

$$\mathbb{E}_{p_Q(V | U_B, \Sigma; \theta)}[V] = \mathbb{E}_{p(V | U, \Sigma)}[w(V) V]. \tag{7}$$

The importance weight $w(V)$ is given by

$$w(V) = \frac{p_Q(V | U_B, \Sigma; \theta)}{p(V | U, \Sigma)} = \frac{p_Q(V | U_B, \Sigma; \theta)}{p_B(V | U_B, \Sigma)} \times \frac{p_B(V | U_B, \Sigma)}{p(V | U, \Sigma)} \tag{8}$$

To simplify the first ratio in (8), we substitute in the expression for $p_Q$ from (4):

$$\frac{p_Q(V | U_B, \Sigma; \theta)}{p_B(V | U_B, \Sigma)} = \frac{1}{Z} \exp\left(-\frac{1}{\lambda}(S(V; \theta)\right) \tag{9}$$

We then simplify the second ratio, as both are Gaussian distributions with the same covariance matrix $\Sigma$:

$$\frac{p_B(V | U_B, \Sigma)}{p(V | U, \Sigma)} \propto \exp\left(-\sum_{k=0}^{K-1} (u_k - u_k^B)^\mathsf{T} \Sigma^{-1} v_k\right), \tag{10}$$

where $u_k$ and $v_k$, $k = 0, 1, \ldots, K-1$ are the mean controls and the sampled controls from $p(V|U, \Sigma)$, respectively. Similarly, $u_k^B$, $k = 0, 1, \ldots, K - 1$ are the mean controls for the base distribution, and they are all $0$ in our case. Substituting (9) and (10) into (8), we have

$$w(V) \propto \exp(-\frac{1}{\lambda}\left( S(V; \theta) + \lambda \sum_{k=0}^{K-1} u_k^{\mathsf{T}} \Sigma^{-1} v_k \right) \tag{11}$$

For each sampled control sequence $V$, we evaluate the expression in (11) and normalize over all samples to obtain $w(V)$.

To summarize, we estimate $\partial \mathcal{L} / \partial \theta$ through sampling:

$$\frac{\partial}{\partial \theta} \mathcal{L}(\theta; \mathcal{D}_t, x_t) \approx \frac{1}{N} \sum_{i=1}^{N} \frac{1}{\lambda} \frac{\partial}{\partial \theta} S(V_i, x_t; \theta) - \frac{1}{M} \sum_{j=1}^{M} \frac{1}{\lambda} w(V_j) \frac{\partial}{\partial \theta} S(V_j, x_t; \theta), \tag{12}$$

where $V_i, i = 1, \ldots, N$ are the control sequences for the expert demonstrations in $\mathcal{D}_t$ and $V_j, j = 1, 2, \ldots, M$ are the sampled control sequences. Equation (12) connects $\partial \mathcal{L} / \partial \theta$ with $\partial S / \partial \theta$. The state cost function $S$ is represented as a shallow neural network, and its derivative can be obtained easily through back-propagation. To evaluate $\frac{\partial}{\partial \theta} S(V_i, x_t; \theta)$, we do not actually use the expert control sequences, as they are unknown. We use the corresponding state trajectories in $\mathcal{D}_t$ directly, as the state cost depends only on the visited states. See equation (3).

Finally, we approximate the optimal mean control sequence according to (7):

$$U = \mathbb{E}_{p_Q(V|U_B, \Sigma, x_t; \theta)}[V] \approx \sum_{j=1}^{M} w(V_j) V_j. \tag{13}$$

The first element in the control sequence $U$ is the chosen control for the current time step $t$. We remove the first element from $U$ and append zero at the end. The new control sequence is then used as the mean for the sampling distribution in the next iteration.

## 3.5 Analysis

Since RHIRL performs local optimization sequentially over many steps, one main concern is error accumulation over time. For example, standard behavior cloning has one-step error $\epsilon$ and cumulative error $O(T^2 \epsilon)$ over $T$ steps, because of covariate shift [30]. The DAgger algorithm reduces the error to $O(T\epsilon)$ by querying the expert repeatedly during online learning [30]. We prove a similar result for RHIRL, which uses offline expert demonstrations only. In iteration $t$ of RHIRL, let $p_E(V_t|x_t)$ be the $K$-step expert demonstration distribution and $p_Q(V_t|U_B, \Sigma, x_t; \theta)$ be the computed $K$-step optimal control distribution for some fixed cost parameters $\theta$. RHIRL minimizes the KL-divergence between these two distributions in each iteration. Let $p_E(x)$ be the state marginal distribution of expert demonstrations and $p_{RHC}(x; \theta)$ be the state marginal distribution of the computed RHC policy under $\theta$ over the entire task duration $T$. Intuitively, we want the control policy under the learned cost to visit states similar to those of expert demonstrations in distribution. In other words, $p_{RHC}(x; \theta)$ and $p_E(x)$ are close.

**Theorem 3.1.** *If* $D_{KL}\big(p_E(V_t|x_t) \parallel p_Q(V_t|U_B, \Sigma, x_t; \theta)\big) < \epsilon$ *for all* $t = 0, 1, \ldots, T - 1$, *then* $D_{TV}(p_E(x), p_{RHC}(x; \theta)) < T\sqrt{\epsilon/2}$.

The theorem says that RHIRL's cumulative error, measured in total variation distance $D_{TV}$ between $p_E(x)$ and $p_{RHC}(x; \theta)$, grows linearly with $T$. The proof consists of three steps. First, in each iteration $t$, if the KL-divergence between two control sequence distributions are bounded by $\epsilon$, so is the KL-divergence between the two corresponding state distributions induced by control sequences. Next, we show that the KL-divergence between the state distributions over two successive time steps are bounded by the same $\epsilon$. Finally, we switch from KL-divergence to total variation distance and apply the triangle inequality to obtain the final result. Note that our local optimization's objective is defined in KL divergence, while the final error bound is in TV distance. We switch the distance measures to get the best from both. Minimizing the KL divergence leads to strong local optimization result, but KL itself is not a proper metric. Therefore, we further bound the KL divergence by TV distance to obtain a proper metric bound for the final result. The full proof is given in Appendix B. Since

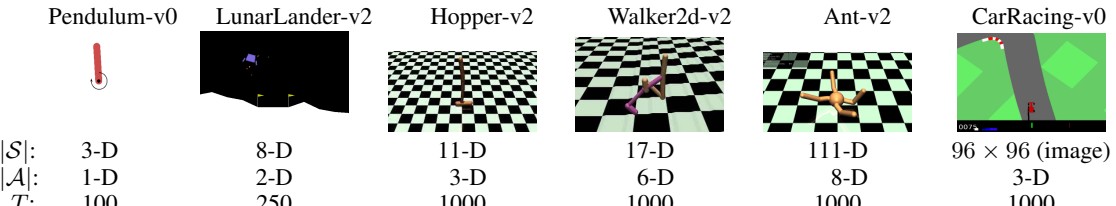

| | Pendulum-v0 | LunarLander-v2 | Hopper-v2 | Walker2d-v2 | Ant-v2 | CarRacing-v0 |
|---|---|---|---|---|---|---|
| $|\mathcal{S}|$: | 3-D | 8-D | 11-D | 17-D | 111-D | $96 \times 96$ (image) |
| $|\mathcal{A}|$: | 1-D | 2-D | 3-D | 6-D | 8-D | 3-D |
| $T$: | 100 | 250 | 1000 | 1000 | 1000 | 1000 |

Figure 2: Benchmark tasks. $|\mathcal{S}|$ and $|\mathcal{A}|$ denote the dimensions of the state space and the action space, respectively. $T$ denotes the task horizon.

RHC performs local optimization in each iteration, we cannot guarantee global optimality. However, the theorem indicates that unlike standard behavior cloning, the cumulative error of RHIRL grows linearly and not quadratically in $T$. This shows one advantage of IRL over behavior cloning from the theoretical angle.

Given a control policy $V$ with the resulting state marginal distribution $p_V(x)$, we can calculate the expected total cost of $V$ by integrating the one-step cost over $p_V$. Now suppose that the one-step cost is bounded. Theorem 3.1 then implies that the regret in total cost, compared with the expert policy, also grows linearly in $T$.

### 3.6 Extension to Stochastic Dynamics

Suppose that the system dynamics is stochastic: $x_{t+1} = f(x_t, v_t, \omega_t)$, where $\omega_t$ is a random variable that models the independent system noise. RHIRL still applies, with modifications. We redefine the total cost functions $\tilde{J}(V, x_0; \theta)$ and $\tilde{S}(V, x_0; \theta)$ by taking expectation over the system noise distribution. When calculating the importance weight $\tilde{w}(V)$ in (11), we sample over the noise distribution to estimate the expected total state cost. Finally, we may need more samples when estimating the gradient in (12), because of the increased variance due to the system noise. The full derivation of the extension to stochastic dynamics is given in Appendix C. The experiments in the current work all have deterministic dynamics. We leave experiments with the extension to future work.

## 4 Experiments

We investigate two main questions. Does RHIRL scale up to high-dimensional continuous control tasks? Does RHIRL learn a robust cost function under noise?

### 4.1 Setup

We compare RHIRL with two leading IRL algorithms, namely AIRL [8] and $f$-IRL [27], and one imitation learning algorithm, GAIL [13]. In particular, $f$-IRL is a recent algorithm that achieves leading performance on high-dimensional control tasks. We use the implementation of AIRL, GAIL, and $f$-IRL from the $f$-IRL's official repository along with the reported hyperparameters [27], whenever possible. We also perform hyperparameter search on a grid to optimize the performance of every method on every task. The specific hyperparameter settings used are reported in Appendix D.2.

Our benchmark set consists of six continuous control tasks (Figure 2) from OpenAI Gym [2], with increasing sizes of state and action spaces. For the most complex task, CarRacing, the input consists of $96 \times 96$ raw images, resulting in an enormous state space that poses great challenges [11]. To our knowledge, RHIRL is the first few IRL algorithms to attempt such a high-dimensional space. For fair comparison, we customize all tasks to the fixed task horizon settings (Figure 2. See Appendix D.2 for details on task parameter settings.

We use WorldModel [11] to generate expert demonstration data for CarRacing and use SAC [12] for the other tasks. We add Gaussian noise to the input controls and collect expert demonstrations at different control noise levels. The covariance of the control noise is *unknown* to all methods, including, in particular, RHIRL.

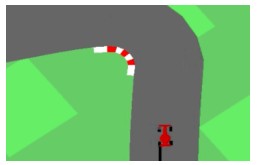 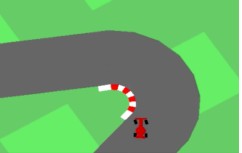 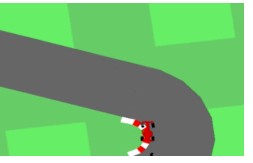 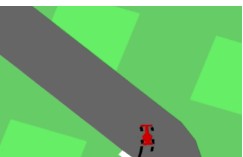

| Drive in the center of the lane. | Adjust the direction at a sharp turn. | Choose the shortest path to make the turn. | Align to the lane center after the turn. |

Figure 3: Driving behaviors learned by RHIRL for CarRacing-v0, with pixel-level raw image input.

To measure the performance of the learned cost function and policy, we score its induced optimal policy using the ground-truth cost function. For ease of comparison with the literature, we use negated cost values, i.e., rewards, in all reported results. Higher values indicate better performance. Each experiment is repeated 10 times to estimate the performance variance.

### 4.2 Scalability

We compare RHIRL with other methods, first in noise-free environments (Figure 4) and then with increasing noise levels (Table 1).

Figure 4 shows the learning curve of each methods in noise-free environments. Overall, RHIRL converges faster and achieves higher return, especially for tasks with higher state space dimensions. This improved performance suggests that the benefit of local optimization adopted by RHIRL outweighs its potential limitations.

Table 1 shows the final performance of all methods at various noise levels. RHIRL clearly outperforms AIRL and GAIL in all experiments. So we focus our discussion on comparison with $f$-IRL. In noise-free environments, RHIRL and

Table 1: Performance comparison of RHIRL and other methods. The performance is reported as the ratio of the learned policy's average return and the expert's average return. The absolute average returns and the standard deviations are reported in Appendix D.4. Negative ratios are clipped to 0. The two numbers under the name of each environment indicate the dimensions of the state space and the action space, respectively.

| | | **No Noise** $\Sigma = 0$ | **Mild Noise** $\Sigma = 0.2$ | **High Noise** $\Sigma = 0.5$ |
|---|---|---|---|---|
| Pendulum 3, 1 | Expert | -154.69 $\pm$ 67.61 | -156.50 $\pm$ 70.72 | -168.54 $\pm$ 80.89 |
| | RHIRL | 1.06 | **1.07** | **1.08** |
| | $f$-IRL | **1.07** | 1.06 | 0.93 |
| | AIRL | 1.05 | 0.94 | 0.91 |
| | GAIL | 0.88 | 0.89 | 0.80 |
| Lunarlander 8, 2 | Expert | 232.00 $\pm$ 86.12 | 222.65 $\pm$ 56.35 | 164.52 $\pm$ 16.79 |
| | RHIRL | **1.05** | **1.04** | **1.20** |
| | $f$-IRL | 0.76 | 0.63 | 0.74 |
| | AIRL | 0.74 | 0.60 | 0.58 |
| | GAIL | 0.72 | 0.56 | 0.60 |
| Hopper 11, 3 | Expert | 3222.48 $\pm$ 390.65 | 3159.72 $\pm$ 520.00 | 2887.72 $\pm$ 483.93 |
| | RHIRL | 0.95 | **0.98** | **0.96** |
| | $f$-IRL | **0.96** | 0.82 | 0.43 |
| | AIRL | 0.01 | 0.01 | 0.01 |
| | GAIL | 0.82 | 0.50 | 0.24 |
| Walker2d 17, 6 | Expert | 4999.47 $\pm$ 55.99 | 4500.43 $\pm$ 114.48 | 3624.48 $\pm$ 95.05 |
| | RHIRL | **0.99** | **0.99** | **0.95** |
| | $f$-IRL | **0.99** | 0.82 | 0.78 |
| | AIRL | 0.00 | 0.00 | 0.00 |
| | GAIL | 0.50 | 0.64 | 0.51 |
| Ant 111, 8 | Expert | 5759.22 $\pm$ 173.57 | 3257.37 $\pm$ 501.95 | 252.62 $\pm$ 91.44 |
| | RHIRL | 0.86 | **0.93** | **0.91** |
| | $f$-IRL | **0.87** | 0.80 | 0.78 |
| | AIRL | 0.17 | 0.33 | 0.00 |
| | GAIL | 0.48 | 0.40 | 0.00 |
| CarRacing $96 \times 96$, 3 | Expert | 903.25 $\pm$ 0.23 | 702.01 $\pm$ 0.3 | 281.12 $\pm$ 0.34 |
| | RHIRL | **0.40** | **0.29** | **0.19** |
| | $f$-IRL | 0.09 | 0.03 | 0.00 |
| | AIRL | 0.00 | 0.00 | 0.00 |
| | GAIL | 0.00 | 0.01 | 0.00 |

$f$-IRL perform comparably on most tasks. On CarRacing, the most challenging task, RHIRL performs much better. RHIRL manages to learn the critical driving behaviors illustrated in Figure 3, despite the high-dimensional image input. However, RHIRL does not manage to learn to drive fast enough. That is the main reason why it under-performs the expert. In comparison, $f$-IRL only learns to follow a straight lane after a large number of environment steps, and still fails to make a sharp turn after $3.0 \times 10^7$ environment steps. In the noisy environments, the advantage of RHIRL over $f$-IRL is more pronounced even on some of the low-dimensional tasks, because RHIRL accounts for the control noise explicitly in the cost function.

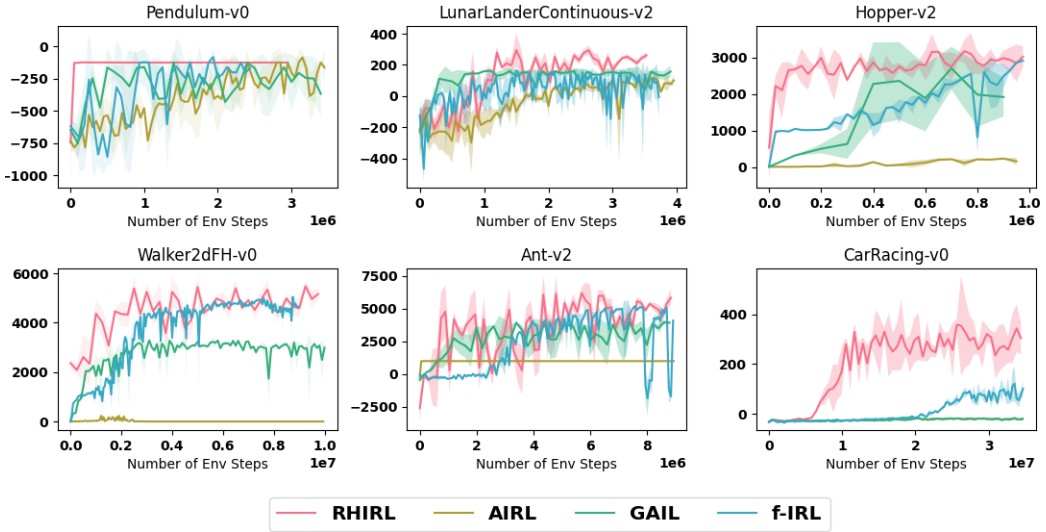

Figure 4: Learning curves for RHIRL and other methods.

## 4.3 Robustness

Next we evaluate RHIRL and other methods for robustness under noise. A robust cost function encodes the expert's true intent. It is expected to yield consistent performance over different noise levels, regardless of noise in expert demonstrations.

For each task, a cost function is learned in a noise-free environment and is then used to re-optimize a policy in the corresponding noisy environments. Specifically for GAIL, since it learns a policy and does not recover the associated cost function, we directly apply the learned policy in noisy environments.

Table 2 shows that noise causes performance degradation in all methods. However, RHIRL is more robust in comparison. For simple tasks, Pendulum and Lunarlander, RHIRL and $f$-IRL perform consistently well across different noise levels, while GAIL and AIRL fail to maintain their good performance, when the noise level increases. For the more challenging tasks, Hopper and Walker, RHIRL's performance degrades mildly, and $f$-IRL suffers more significant performance degradation. It is worth noting that the expert demonstrations used to train the transferred cost function are from the perfect system. Therefore, some expert actions and states may no longer be optimal or feasible in a highly noisy environment.

Table 2: Robustness of RHIRL and other methods under noise. The performance is measured as the ratio between the average return of an re-optimized policy in a noisy environment and the expert's average return in the corresponding noise-free environment. The absolute average returns and the standard deviations are reported in Appendix D.4. The negative ratios are clipped to 0.

| | | Noise Level $\Sigma$ | | |
| --- | --- | --- | --- | --- |
| | | 0.0 | 0.2 | 0.5 |
| Pendulum | Expert | -154.69 ± 67.61 | – | – |
| 3, 1 | RHIRL | 1.06 | **1.07** | **1.06** |
| | $f$-IRL | **1.08** | 0.90 | 0.85 |
| | AIRL | 1.05 | 0.79 | 0.67 |
| | GAIL | 0.88 | 0.71 | 0.62 |
| LunarLander | Expert | 232.00 ± 86.12 | – | – |
| 8, 2 | RHIRL | **1.05** | **0.89** | **0.76** |
| | $f$-IRL | 0.76 | 0.53 | 0.44 |
| | AIRL | 0.74 | 0.14 | 0.10 |
| | GAIL | 0.72 | 0.44 | 0.34 |
| Hopper | Expert | 3222.48 ± 390.65 | – | – |
| 11, 3 | RHIRL | 0.95 | **0.80** | **0.67** |
| | $f$-IRL | **0.96** | 0.65 | 0.62 |
| | AIRL | 0.01 | 0.01 | 0.00 |
| | GAIL | 0.82 | 0.07 | 0.06 |
| Walker | Expert | 4999.47 ± 55.99 | – | – |
| 17, 6 | RHIRL | **0.99** | **0.80** | **0.69** |
| | $f$-IRL | **0.99** | 0.60 | 0.22 |
| | AIRL | 0.00 | 0.28 | 0.36 |
| | GAIL | 0.50 | 0.02 | 0.02 |
| Ant | Expert | 5759.22 ± 173.57 | – | – |
| 111, 8 | RHIRL | **0.86** | **0.55** | **0.15** |
| | $f$-IRL | 0.87 | 0.35 | 0.08 |
| | AIRL | 0.17 | 0.15 | 0.00 |
| | GAIL | 0.48 | 0.00 | 0.00 |
| CarRacing | Expert | 903.25 ± 0.23 | – | – |
| 96 × 96, 3 | RHIRL | **0.40** | **0.29** | **0.12** |
| | $f$-IRL | 0.09 | 0.02 | 0.00 |
| | AIRL | 0.00 | – | – |
| | GAIL | 0.00 | – | – |

Moreover, the cost function trained in the perfect system cannot reason about the long-term consequences of an action in a high noise environment. Therefore, it is challenging for the learned cost function to be robust to a highly noisy environment, as capturing the true intention of the expert is difficult.

### 4.4 Effect of Receding Horizon $K$

RHIRL uses the receding horizon $K$ to trade off optimality and efficiency. We hope to ablate the effect of K on Hopper-v2 to show how different $K$s affect the final performance and sample complexity. The task horizon for Hopper-v2 is 1000 steps, i.e. $T = 1000$. We run RHIRL with the receding horizon $K \in \{5, 20, 100\}$. The results are illustrated in Figure 5. When $K$ is small, RHIRL improves its performance quickly but converges to the suboptimal solution. For $K = 5$, RHIRL's performance shoots up after the first few iterations to 1000, then it quickly converges to a final score of 1100. When $K$ increases, though the performance improves slightly slower than $K = 5$, it can continue to learn and reach a score of 3071.68. At $K = 20$, it takes fewer than $1e6$ env steps to stabilize to a score greater than 3000. However, when $K$

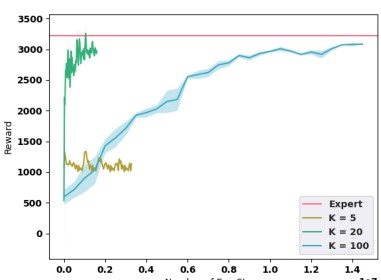

Figure 5: The effect of receding horizon $K$ on the performance of RHIRL on the Hopper-v2 task.

is too large, the learning becomes much slower. When $K = 100$, it takes more than $1e7$ env steps to stabilize to a score larger than 3000, which is 10 times more than when $K = 20$. On the other hand, $K = 100$ can achieve a final score of 3083, which is slightly more than that of $K = 20$. This ablation study shows that our receding horizon $K$ can tradeoff optimality and efficiency: using a smaller $K$ allows us to learn faster at the expense of a sub-optimal solution, while using a large $K$ may make the learning inefficient. Seeking a suitable $K$ can balance the requirement for optimality and efficiency.

## 5   Conclusion

RHIRL is a scalable and robust IRL algorithm for high-dimensional, noisy, continuous systems. Our experiments show that RHIRL outperforms several leading IRL algorithms on multiple benchmark tasks, especially when expert demonstrations are noisy.

RHIRL's choice of local rather than global optimization is an important issue that deserves further investigation. Overall, we view this as an interesting trade-off between scalability and optimality. While this trade-off is well known in reinforcement learning, optimal control, and general optimization problems, it is mostly unexplored in IRL. Further, local optimization may tie the learned cost with the optimizer. It would be interesting to examine whether the learned cost transfers easily to other domains with different optimizers. We are keen to investigate these important issues and their implications to IRL as our next step.

**Acknowledgments**   This research is supported in part by the Agency of Science, Technology and Research, Singapore, through the National Robotics Program (Grant No. 192 25 00054), the National Research Foundation (NRF), Singapore and DSO National Laboratories under the AI Singapore Program (AISG Award No: AISG2-RP-2020-016), and the National University of Singapore (AcRF grant A-0008071-00-00). Any opinions, findings and conclusions or recommendations expressed in this material are those of the authors and do not reflect the views of NRF Singapore.

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
