# OpenReview forum: "Receding Horizon Inverse Reinforcement Learning"
_NeurIPS.cc/2022/Conference — NeurIPS 2022 Accept_

### Official Review · Reviewer_VXKi · 2022-07-09

**Rating:** 3
**Confidence:** 2
**Soundness:** 3 good
**Presentation:** 2 fair
**Contribution:** 2 fair

**Summary:**

This paper proposes a method for inverse RL based on optimizing short-horizon, open-loop action sequences. Similar to MPPI, the proposed method learns a policy by sampling a bunch of these open-loop action sequences, weights each by its (learned) reward function, and then updates the mean control sequence. After updating the policy, the proposed method updates the reward function using gradient descent. Empirically, the proposed method outperforms three imitation learning baselines on 6 tasks, and ablation experiments show that it is relatively robust to noise in the demonstrations.

**Questions:**

Questions:
*  Most inverse RL and imitation learning methods can be simply described as a combination of (1) an inverse RL objective (e.g., distribution matching for MaxEnt IRL, a max-margin objective for max margin IRL) and (2) and RL algorithm (e.g., MPC, MPPI, LQR, SAC). Is the proposed method equivalent to doing MaxEnt IRL with MPPI, with a special parametrization of the cost function (Eq. 2)?
* One of the central questions raised by the paper was the tradeoff vs local vs global methods. I didn't fully understand this point, and was hoping that the authors could elaborate a bit. Is the trade-off caused by using $K$ step planning instead of planning over infinite-length horizons? Or is it caused by planning over open loop sequences rather than closed loop sequences (i.e., contingency planning)?
* Does the method require reset access to the simulator?
* How does the proposed method compare to more recent+competitive baselines, such as (well-tuned) behavioral cloning and DAC [1]?
* L37 "trades off optimality" -- Does this mean that the proposed method doesn't converge to the reward-maximizing policy?
* Do the baselines also factor the reward function as in Eq. 2?

**Summary**: Overall, I think the paper is generally strong. The proposed method seems theoretically sound, and empirically seems to outperform some baselines. My current score reflects uncertainty about three main aspects of the paper:
1. I'm not clear about the difference between the proposed method and MaxEnt IRL + MPPI
2. I'm not sure whether the proposed method requires resetting the environment, in ways that the alternative approaches don't
3. It's not clear how the proposed method would compare to more recent/competitive imitation learning methods.

If these concerns are addressed, I would definitely consider increasing my score.

Minor comments
* L16 "on line" --> "online
* L19 "etc" -- Don't use "etc" in technical writing.
* L21 "in practice" -- Cite.
* L30 -- L34 -- I didn't understand this part of this paragraph.
* Fig 1 -- I didn't understand this figure. It might be easier to visualize just one initial state for each example.
* L67 "challenges in scalability" -- I'd recommend elaborating on this point.
* L70 "major obstacle" -- Cite.
* L96 "full proof" -- Proof of what?
* Eq 2 -- Where is $J$ used in the subsequent derivation?
* Fig 4 -- I didn't find this figure very informative, and recommend that it be replaced by a figure that explains the mechanics of the method (e.g., visualize a bunch of rollouts, visualize their rewards, visualize how this changes the mean control and the reward function)


[1] https://arxiv.org/abs/1809.02925

**Limitations:**

Yes, the second paragraph of the conclusion has a great discussion of the trade-off between scalability vs optimality induced by the proposed method.

**Strengths And Weaknesses:**

Strengths
* The proposed method is relatively simple.
* I appreciate that the paper explains how the paper fits into the larger body of prior work by relating the tradeoffs made by shorter horizon planning.
* I appreciate that the paper shows results on the image-based car-racing task, showing that the method isn't limited to low-dimensional settings.
* The ablation experiments are great for studying the robustness of the method.

Weaknesses
* I'm a bit unsure whether the proposed method requires more assumptions than prior work. For example, the trajectory sampling in L6 of Alg 1 seems to require reset access to the simulator, which most prior methods avoid.
* The proposed method seems fairly similar to prior work on imitation learning [1,2, 3].

[1] https://arxiv.org/pdf/1206.4617.pdf

[2] http://proceedings.mlr.press/v48/finn16.pdf

[3] https://arxiv.org/pdf/2201.06539.pdf

---

> ### Author Response · Authors · 2022-08-02
> **Response to Reviewer VXKi**
>
> Thank you for the suggestions and questions. Let’s start with the 3 main ones listed under Summary.
>
> Q1: The difference between the proposed method and MaxEnt IRL + MPPI.
>
> RHIRL is not simply a combination of MaxEntIRL’s objective with MPPI as the forward solver. The primary difference is the optimization objective. RHIRL matches the induced local trajectory distribution over the chosen horizon K, with expert demonstrations. MaxEntIRL (Ziebart et. al, 2008) matches the global trajectory distributions over the full horizon. This is the key difference that, when combined with the sampling-based MPPI for optimization, leads to the scalability of RHIRL to high-dimensional spaces and long-horizon tasks.
>
> RHIRL and MaxEntIRL become similar in their objective in the special case when the transition function is deterministic and the receding planning horizon K is the same as the task horizon T.
>
> The scalability of RHIRL relies on three key ingredients:
> - A structured representation of the task cost
> - Local, rather than global, match of trajectory distributions
> - MPPI as the optimizer, over a chosen finite horizon
>
> All three are required to work TOGETHER, to enable RHIRL to scale up and handle high-dimensional, long-horizon tasks.
>
> It is indeed true that most inverse RL and imitation learning methods are a combination of (1) an inverse RL objective and (2) and RL algorithm. The Introduction of our paper alludes to this issue and refers to it as the “double-loop structure. It’s the specific choices for the IRL objective, the optimizer, as well as the representation of the task (in our particular case), that differentiate their scalability, robustness, …
>
> Q2: Resettable simulator.
>
> This is a great point for us to clarify more! RHIRL assumes a resettable simulator, a common setting used in earlier work, including REIRL, GCL, AIRL, and DAC. Other earlier work, including, particularly, MaxEntIRL assumes a distributional model of dynamics, generally a stronger assumption than that a black-box simulator.
>
> Q3: It's unclear how the proposed method would compare to more recent/competitive imitation learning methods.
>
> BC is simpler but generally has higher sample complexity and suffers from covariant shift. “Tuning” cannot resolve these fundamental issues.
>
> DAC(2018) belongs to the class of AIL methods that aims to improve sample efficiency and reward bias. DAC addresses the sample efficiency using off-policy training and resolves the reward bias by separately learning a reward for the absorbing states. However, it still operates on the global state-action/state marginal distribution, therefore does not resolve the scalability issue for tasks in high-dim long horizon IRL tasks. Unfortunately, the DAC paper does not report absolute performance scores, and we cannot directly compare.
>
> We have compared with f-IRL(2020), a more recent and competitive baseline that aims to improve IRL's sample complexity. Our experimental results show that we outperform f-IRL in several benchmarks.
>
> Now we answer the remaining questions.
>
> Q4. What are the cause and implications of the tradeoff between scalability and optimality have?
>
> This trade-off is the direct consequence of planning over a K-step horizon instead of the full task horizon. RHIRL saves computation by optimizing the controls over a short horizon. Hence, the optimality of the learned reward function is only assured in this local region.
>
> Despite the trade-off, our theoretical analysis (Theorem 3.1) establishes an error bound that measures the deviation between our induced trajectory with the expert. It shows the error grows linearly in the task horizon T. The experiment results also show that the trajectories induced by RHIRL are evaluated to high scores under the ground-truth reward functions. All these indicate that RHIRL induces approximately optimal trajectories.
>
> For clarification, while RHIRL uses MPPI to perform open-loop control optimization, it closes the loop through re-planning. This issue, however, has no bearing on the trade-off between scalability and optimality.
>
> Q5. Do the baselines factor the reward function as in Eq. 2?
>
> No. We follow the reward function parameterization in their original paper to reproduce the reported performance.  To our best knowledge, we are not aware of earlier IRL methods that take advantage of our structured reward representation.
>
> Q6. Comparison with prior works on imitation learning [1,2, 3].
>
> The first two works are contrasted in response to Reviewer xWf5, Q1.3.
>
> [3] is an MPC framework that leverages on MEIRL to learn a costmap. Its reward function is a linearly weighted combination of (non-linear) state features for simple IRL gradient derivation. In contrast, our reward function is a sum of two components: an arbitrary nonlinear state component and a quadratic control component.  Our choice allows RHIRL to handle a more flexible and broader class of reward functions nonlinear in state features and robust to noise.

---

> > ### Comment · Reviewer_VXKi · 2022-08-02
> > **Reviewer reseponse**
> >
> > Thanks to the authors for the detailed response! I appreciate all the time they put into responding to the questions and concerns raised in the review. I have a few follow-up questions and clarifications
> >
> > > The difference between the proposed method and MaxEnt IRL + MPPI.
> >
> > To help me understand the difference, would it be possible to provide a precise comparison of the MaxEnt IRL objective next to the RHIRL objective, using as much of the same notation as possible?
> >
> > > RHIRL assumes a resettable simulator, a common setting used in earlier work, including REIRL, GCL, AIRL, and DAC
> >
> > I'm a bit confused by this. My understanding of AIRL and DAC was that the simulator was reset periodically, but that the algorithm couldn't reset the agent to some specific state of interest (e.g., to try out a different action sequence). Does RHIRL assume that the algorithm can reset the agent to any specific state of interest?
> >
> > > Unfortunately, the DAC paper does not report absolute performance scores, and we cannot directly compare.
> >
> > Would it be possible to run the DAC algorithm in the same settings as RHIRL? The GitHub repo [1] says that training takes just 1-2 hr.
> >
> > > Do the baselines factor the reward function as in Eq. 2?
> >
> > Would it be possible to ablate this design decision? The paper seems to argue that the local vs global design decision is the important one, but the factorization of the reward function seems like a potential confounding factor.
> >
> > [1] https://github.com/google-research/google-research/blob/master/dac/README.md

---

> > > ### Author Response · Authors · 2022-08-04
> > > **Response to Reviewer VXKi (part 3)**
> > >
> > > *Q3. Would it be possible to run the DAC algorithm in the same settings as RHIRL? The GitHub repo [1] says that training takes just 1-2 hr.*
> > >
> > > We are currently adapting our settings and the expert demonstrations to DAC. The expert demonstrations provided by the official DAC github repo contain the ground-truth reward for every transition. Our expert demonstration data does not have this information and DAC algorithm presented in the paper does not seem to use it, hence the reward entry of expert demonstrations is assigned to 0 to match with the required expert demonstration's format. We shall report the performance on the mujoco tasks shortly. We need more time to run DAC on CarRacing due to its high-dimensional pixel-wise image input. We’ll update you on the progress and present the new result as soon as possible.
> > >
> > > *Q4. Would it be possible to ablate this design decision? The paper seems to argue that the local vs global design decision is the important one, but the factorization of the reward function seems like a potential confounding factor.*
> > >
> > > Thank you for the suggestion! To ablate on this design decision, we plan to only use the state-dependent cost for optimization in a noise-free environment. This can be done by setting the inverse temperature $\lambda$ to 0. This reduces our linearly separable cost function to a state-only cost, the same as how AIRL, f-IRL parameterize the reward function. We are currently working on this and will report the result shortly.

---

> > > ### Author Response · Authors · 2022-08-04
> > > **Response to Reviewer VXKi (part 2)**
> > >
> > > *Q2. I'm a bit confused by this. My understanding of AIRL and DAC was that the simulator was reset periodically, but that the algorithm couldn't reset the agent to some specific state of interest (e.g., to try out a different action sequence). Does RHIRL assume that the algorithm can reset the agent to any specific state of interest?*
> > >
> > > This is a great point for us to clarify more!
> > > - RHIRL requires our simulator to reset the agent to a specific state of interest. MPPI is a sampling-based controller that requires the simulator to reset to the current state of interest to optimize.
> > > - AIRL and DAC need to reset to the distribution of initial states periodically, while RHIRL uses the simulator to reset to every state along the trajectory to perform receding horizon learning.
> > >
> > > They differ in the number of reset operations. We believe that two simulators that differ in the number of states that can be reset to are not significantly different. Shall we understand a bit more here, for the reviewer's concerns on this reset capability and frequency of the simulator, is it more on its computational cost or its implementation difficulty? Thank you!

---

> > > > ### Comment · Reviewer_VXKi · 2022-08-04
> > > > **Reviewer response**
> > > >
> > > > Thanks for clarifying this assumption!
> > > >
> > > > > Shall we understand a bit more here, for the reviewer's concerns on this reset capability and frequency of the simulator, is it more on its computational cost or its implementation difficulty?
> > > > A few concerns here are:
> > > > * It seems like the baselines in the current paper make fewer assumptions than the proposed method, so the comparison may be a bit unfair.
> > > > * In practice, these sorts of resets seem like they may be harder to implement than initial state resets. For example, in an autonomous driving application, it seems easier to return the car to the center of the lane (the initial state distribution) than to a more specific traffic configuration.

---

> > > > > ### Author Response · Authors · 2022-08-06
> > > > > **response to reviewer VXKi**
> > > > >
> > > > > Thank you for clarifying your concern about the resettable simulator! We would like to express our view on the resettable simulator here.
> > > > >
> > > > > It is unnatural to assume a simulator resettable to only a subset of states. What is the underlying algorithmic resetting mechanism to make it so? For autonomous driving in practice, such a resetting assumption would be even more problematic. The vehicle must drive perfectly along the center of the lane all the time. What if it deviates, which happens all the time? What if the vehicle does not start at the center of the lane from the beginning?  In general, we want to assume that the simulator is resettable to any state.
> > > > >
> > > > > Our assumption of a resettable simulator is stated clearly upfront. The comparison is fair in the sense that the same experimental setup is given to all the methods. The results support our conclusion, under the explicitly stated assumption.

---

> > > > > > ### Comment · Reviewer_VXKi · 2022-08-06
> > > > > > **Reviewer response**
> > > > > >
> > > > > > Thanks for the response!
> > > > > >
> > > > > > I remain concerned about this point. One revision that would help address this concern would be to revise the Overview (section 3.1) to state this assumption explicitly (the word "reset" currently doesn't appear in the paper) and to discuss how this assumption is different from prior work. For example, this assumption seems different from the standard MDP model, which assumes that resets happen by sampling from an initial state distribution, not by resetting to algorithm-specified states. It would also be great to discuss how common this assumption is. For example, the MaxEnt IRL paper doesn't seem to make this assumption, but I think the MPPI paper does. Including such a comparison would help readers situate the proposed method relative to prior methods that do/don't make this additional assumption.

---

> > > > > > > ### Author Response · Authors · 2022-08-08
> > > > > > > **Response to Reviewer VXKi**
> > > > > > >
> > > > > > > This is a great suggestion! We have revised the overview (section 3.1, line 122- 126) to highlight that the receding horizon performs local optimization successively, this implies that we need a simulator that supports reset to a specific state. We have compared with the prior IRL works based on the RL optimizer and on the optimal controllers, accounting for their different assumptions on the simulators. This will definitely help the reader to better position and understand the context of our work! Thank you!
> > > > > > >
> > > > > > > The revised version is as follows:
> > > > > > >
> > > > > > > *'Earlier IRL work \cite{AIRL, GAIL, REIRL, GCL, GAN-GCL, DAC}  requires the simulator to reset to the states in the initial state distribution. To perform our receding horizon optimization, RHIRL requires a simulator to reset to the initial state $x_t$ for the local control sequence optimization at each time step $t = 0, \ldots, T-1$. This is a common setting for the work, such as \cite{MPPI, RobustMPC, CostMPC}, from the optimal control community.'*

---

> > > ### Author Response · Authors · 2022-08-04
> > > **Response to Reviewer VXKi (part 1)**
> > >
> > > Thank you for follow-up questions! We would like to clarify some of the concerns and report on our additional experiment plans.
> > >
> > >  Q1. To help me understand the difference, would it be possible to provide a precise comparison of the MaxEnt IRL objective next to the RHIRL objective, using as much of the same notation as possible?
> > >
> > > This is a great suggestion to distinguish our objective from MaxEntIRL clearly!
> > >
> > > Consider a task with length $T$ time steps.  To abstract away the different design choices for the reward parameterization, let's use $\theta$ to represent the parameters of the unknown reward function generically. We are given a set of expert demonstrations $\mathcal{D}$ sampled from an expert policy.
> > >
> > > MaxEntropy IRL optimizes the entropy over the trajectory distribution. A trajectory is a sequence of states and
> > > controls of length $T$, i.e. $\tau = (x_0, u_0, x_1, \ldots, u_{T-2}, x_{T-1})$. MaxEntropy IRL maximizes the likelihood of the expert demonstrations $\mathcal{D}$ under the optimal trajectory distribution. MaxEntIRL’s objective function is as follows:
> > > $\theta^* = \underset{\theta}{\arg\max} \sum_{\tau\sim \mathcal{D}} \log p(\tau|\theta)$,
> > > where $p(\tau|\theta)$ is the optimal trajectory distribution under the maximum entropy principle:$$p(\tau^*|\theta) = \frac{1}{Z(\theta)}exp(R(\tau^*; \theta))  (1).$$ We highlight that $Z(\theta)$ is the partition function given by $Z(\theta) = \int_{\tau}R(\tau; \theta) d\tau$ and $R(\tau; \theta)$ is the cumulative reward of the trajectory .
> > >
> > > In contrast, RHIRL optimizes over the local control sequence space.  A control sequence $V = (u_0, u_1, \ldots, u_{K-2})$ has K-1 consecutive controls. We minimize the KL divergence between the expert demonstration and our control sequence distribution starting at every iteration t:
> > > $\theta^* = \underset{\theta}{\arg\min} D_{KL} \bigl(p_{ E}(V|x_t) \parallel p(V| \Sigma,x_t;\theta)\bigr)$,
> > > where $p_{ E}(V|x_t)$ is the expert demonstrated local control sequence distribution, while $p(V| \Sigma,x_t;\theta)$ is the optimal local control sequence distribution based on the minimum “free energy” principle:
> > > $$p(V^*| \Sigma, x_t; \theta) = \frac{1}{Z(\theta)}\{p_{B}(V^*|U_B, \Sigma) \exp\bigl(-\frac{1}{\lambda} S(V^*, x_t;\theta)\bigr)}(2)$$
> > > where $Z (\theta)$ is our partition function and is given by  $Z(\theta) = \int_{V}\,{p_{B}(V^*|U_B, \Sigma) \exp\bigl(-\frac{1}{\lambda} S(V^*, x_t;\theta)\bigr)}dV$ and $S(V, x_t;\theta)$ is the total cost of a control sequence of length K-1.
> > >
> > > At first glance, our expression for the optimal control sequence distribution in equation (2) looks similar to that of MaxEntIRL in equation (1). However, they differ significantly in their partition functions. For MaxEntIRL, the partition function $Z$ is computed assuming that every trajectory of length T through the MDP with the same value is equally likely. This integral becomes computationally intractable in high-dimensional and long horizon problems due to the curse of dimensionality. In contrast, the control sequence distribution in the partition of equation (2) is generated by the system's dynamics by executing short horizon controls of length K. This partition function only covers the physically reachable local region. Essentially, our optimal control sequence $V^*$ is only compared against control sequences in its local vicinity rather than considering the entire space of trajectories uniformly. Locality in our setting is determined by the length of the receding horizon K. We can control the sample complexity to estimate this partition function by the receding horizon K. This is why our approach is more suitable in high-dimensional and long horizon settings.
> > >
> > > Another subtle difference is that our partition has an additional Gaussian ‘’base’’ distribution term $p_{B}(V^*|U_B, \Sigma)$ in our partition function. This term allows us to inject the control prior to the sampling process if we have any.
> > >
> > > Moreover, minimizing the KL divergence over the control sequence space, coupled with the structural parameterization of the cost function, enables us to derive a simple gradient to update our cost function. A unique benefit is that,  though we optimize over the control sequence distribution, our gradient computation only needs the state sequences of the expert. This frees us from collecting the expert control sequences, which can be tedious and difficult sometimes when the controls are unobservable. The details of our gradient derivation are presented in section 3.4.
> > >
> > > We hope this explanation can help to clarify some of your doubts. Let us know if you have further questions!

---

> > > > ### Comment · Reviewer_VXKi · 2022-08-04
> > > > **Reviewer response**
> > > >
> > > > Thank you for further clarifying the connections!
> > > >
> > > > Pattern matching Equation 1 and 2, it seems like RHIRL is a special case of MaxEnt IRL where the reward $R(\tau)$ from Eq. 1 (1) only depends on the actions, not the states; and (2) is zero for all states and actions where $r > k-1$. Is this summary correct?

---

> > > > > ### Author Response · Authors · 2022-08-06
> > > > > **Clarification on Reviewer's response**
> > > > >
> > > > > Thank you for your question! We would like to point out two issues from this summary:
> > > > >
> > > > > 1. $R(\tau)$ in equation (1) depends on both states and the actions. We follow the convention for the RL literature, $R(\tau)$ denotes the total reward of a trajectory $\tau = (x_0, u_0, x_1, \ldots, u_{T-2}, x_{T-1})$, that is $R(\tau) = \sum_{t=0}^{T-1} r(x_t, u_t;\theta)$. $R(\tau)$ depends on both states and actions; this is consistent with many works based on MaxEntropy IRL’s objective function, such as Guided Cost Learning (GCL) and Relative Entropy IRL(REIRL).
> > > > >
> > > > > 2. Do you mean “and (2) is zero for all states and actions where $t > K-1$”? I interpret this question as: since RHIRL performs only local optimization of length $K$, does it mean that the state cost and control cost beyond $K$ are all zero? My answer below will be based on my interpretation.
> > > > >
> > > > > The state cost learned by RHIRL is not zero beyond length $K$.  We perform the local optimization over a window of $K$ successive time steps, eventually, the window slides over the entire space of length $T$. Therefore, the state cost along the expert demonstrations of length $T$ will all be updated eventually. I believe the confusion lies in the receding horizon nature of RHIRL: the objective function stated above and optimal control sequence distribution in equation (2) are both local measures of length K; however, we optimize our local objective function for each time step $t=1,.., T-1$. Therefore, eventually, the state cost along the entire trajectory of length $T$ will be learned. More especially,  for a single time step, we only update the state cost of the local region that can be reached by a control sequence of length $K$. Then we execute one control from the local optimization solution and transit to the new state. We continue to optimize the state cost in the local vicinity of this new state. This process repeats and eventually, the state cost along the entire trajectory of length $T$ will be updated.
> > > > >
> > > > > We are not a special case of MaxEntIRL. RHIRL can choose the size of the region to optimize over using the receding horizon $K$. In contrast, MaxEntIRL can only optimize over the global trajectory space of fixed length $T$. This means that if the task horizon is short, RHIRL can take K=T to optimize over the entire space, the same as the MaxEnt IRL.  However, when it comes to high-dimensional and long horizon tasks, where optimizing for the entire trajectory of length T is computationally intractable, MaxEntIRL’s objective function will fail due to the ``curse of dimensionality’’. In contrast, RHIRL can still optimize the local problem and “stitch” the local solution together for a global solution.

---

> > > > > > ### Comment · Reviewer_VXKi · 2022-08-06
> > > > > > **Reviewer response**
> > > > > >
> > > > > > Thanks for the further clarifications.
> > > > > >
> > > > > > > $R(\tau)$ depends on both states and actions.
> > > > > >
> > > > > > OK, this makes sense, and resolves my concerns here. It might be worth clarifying this in the paper. E.g., the notation $S(V, x_t; \theta)$ in Eq. 4 seems to imply that the cumulative cost function depends on a _sequence_ of actions, but only on the first observation $x_t$.
> > > > > >
> > > > > > > I believe the confusion lies in the receding horizon nature of RHIRL
> > > > > >
> > > > > > Thanks for clarifying this! I mistakenly assumed that MaxEnt IRL, Relative Entropy IRL, and GCL would all do something similar, performing gradient updates on all trajectory _snippets_ $\tau_{t:T}$ for $0 \le t < T$. However, reviewing these prior works, it does seem like they operate on the original trajectories, without considering sub-trajectories.
> > > > > >
> > > > > > Looking at some more recent work [1, Eq. 3], it seems like the MaxEnt IRL objective can be written in terms of the state occupancy measure. It seems like such a formulation would have an effect similar to receding horizon control, in that the objective takes gradient steps not just starting at the initial state, but also all states visited thereafter.  **Is the "receding horizon" part of RHIRL functionally different from state-occupancy formulations of IRL?**
> > > > > >
> > > > > > In practice, it seems like adding a discount factor would have a similar effect to truncating the rollouts. Precisely, discounting is exactly equivalent to truncating the rollouts at a _random_ length, determined by a geometric distribution. **Does truncating the rollouts at a fixed length, as done by RHIRL, have a functionally different effect than truncating them at a random length, as would be motivated from the discounted sum of returns?**
> > > > > >
> > > > > >
> > > > > > [1] Garg, Divyansh, et al. "IQ-Learn: Inverse soft-Q Learning for Imitation." Advances in Neural Information Processing Systems 34 (2021): 4028-4039.

---

> > > > > > > ### Author Response · Authors · 2022-08-08
> > > > > > > **Response to reviewer VXKi**
> > > > > > >
> > > > > > > RHIRL’s objective function is different from matching state occupancy measure.
> > > > > > >
> > > > > > > Matching the state occupancy measure is still a global optimization method first proposed by GAIL [1, eq. 8]. In fact, MaxEntropy IRL is a dual of the state occupancy measure problem. The state occupancy measure $\rho(s, a)$ is the marginal state-action pair distribution over the entire space. More specifically, for a policy $\pi \in \Pi $, the precise mathematical definition for its occupancy measure $\rho_{\pi} : S \times A \to \mathbb{R}$ as
> > > > > > > $\rho_{\pi}(s, a) = \pi(a|s) \sum_{t=0}^{\infty} \gamma^{t} p(s_t = s|\pi) $ (1).
> > > > > > >
> > > > > > > **Matching the state occupancy measure still falls under the umbrella of the global optimization methods, since its objective function requires estimating the global state occupancy measure for the expert demonstrations $\rho_E $ and the current policy $\rho_{\pi}$.** This contrasts sharply with RHIRL, which only matches the local control sequence distribution.
> > > > > > >
> > > > > > > The root difference between RHIRL and the baselines lies in the objective functions; we optimize locally at each time, while the baselines optimize over a global distribution measure, even with a smaller discount factor. Truncating the rollouts and using discounted factor are just two different ways to achieve the effect of accounting returns from a short horizon. Themselves do not make a huge different in the scalability of the IRL algorithms. **The key difference is in the algorithmic choice of the optimizer - whether the optimizer operates in a local or global region.** RHIRL uses the forward search over a short horizon for each time step; through this, we control our computation complexity. In contrast, for the baselines, even with a smaller discount factor,  they are still learning the global value function to derive the policy.  Having a smaller discount factor may speed up their convergence, but still does not fundamentally change its nature of global optimization.

---

> > > > > > > > ### Comment · Reviewer_VXKi · 2022-08-09
> > > > > > > > **Reviewer response**
> > > > > > > >
> > > > > > > > Thanks for continuing the discussion!
> > > > > > > >
> > > > > > > > > state occupancy measure
> > > > > > > >
> > > > > > > > Sorry about the confusion here; I did a poor job explaining the question. By "state-occupancy formulations of IRL," I meant objective functions that take an expectation over the state occupancy measure (e.g., [1, Eq. 3]). I didn't mean methods that minimize a divergence measure between state occupancy measures (though, of course, these two types of methods are very closely related).
> > > > > > > >
> > > > > > > > > The root difference ... lies in the objective functions; we optimize locally at each time, ...
> > > > > > > >
> > > > > > > > I'm still a bit confused about this difference, because the explanation seems to conflate two different levels of analysis: the objective function, and the optimization algorithm. My current mental model of RHIRL is that it is effectively optimizing the same objective as MaxEnt IRL (perhaps with smaller discount factor than usual), but using a very different optimization algorithm. If this is correct, it seems like the empirical results in the paper remain strong (though I really do think it's important to try running the baselines with a small discount factor, at least for the next version), while the rhetorical arguments about global/local optimization might not be correct.
> > > > > > > >
> > > > > > > > [1] Garg, Divyansh, et al. "IQ-Learn: Inverse soft-Q Learning for Imitation." Advances in Neural Information Processing Systems 34 (2021): 4028-4039.

---

> > > > > > > > > ### Author Response · Authors · 2022-08-09
> > > > > > > > > **Response to review VXKi**
> > > > > > > > >
> > > > > > > > > Thank you for further clarifying your questions. We would like to respond to the following two issues.
> > > > > > > > >
> > > > > > > > > State occupancy measure:
> > > > > > > > >
> > > > > > > > > Thank you for the clarification. I would like to point out that the state occupancy measures, by its definition, is the (global) marginal state-action pair distribution over the entire space. The state occupancy measure is not for the local state visitation, hence it does not support an easy way to perform local optimization. Matching state occupancy measure is completely different from RHIRL, where we only approximate a local region of states. This enables us to carry out a sequence of local optimization to resolve the “curse of dimensionality”. In short, RHIRL is not MaxEntIRL + MPPI, because of this key issue of optimizing over global vs local feature distributions.
> > > > > > > > >
> > > > > > > > > RHIRL is not MaxEntIRL and MPPI:
> > > > > > > > >
> > > > > > > > > MaxEntIRL optimizes over a global distribution measure of the policy, be it trajectory distribution, or state marginal distribution. It does not easily admit a local optimizer to successively solve short horizon problems and stitch them together in a receding horizon manner. Using a smaller discount factor does not stop MaxtEntrIRL’s objective from optimizing over the global distribution. It is the receding horizon design with local optimization that contributes to scalability.

---

> > > ### Author Response · Authors · 2022-08-06
> > > **Additional results on DAC**
> > >
> > > We have adapted our settings on all six tasks to the official DAC repository. We follow the official implementation to run 1e6 training steps for each task; the training time for each task ranges from 12 to 24 hrs. The final performance scores after training are reported below.
> > >
> > > |             | No Noise $\Sigma = 0$ | Mild Noise $\Sigma = 0.2$ | High Noise $\Sigma = 0.5$ |
> > > |-------------|-----------------------|--------------------------|---------------------------|
> > > | Pendulum    | -166.81 $\pm$ 67.83   | - 264.50 $\pm$ 79.98     | -348.57 $\pm$ 143.21      |
> > > | LunarLander | 160.44 $\pm$ 45.01    | -0.25 $\pm$ 2.18         | -4.05 $\pm$ 3.19          |
> > > | Hopper      | 3291.91 $\pm$ 167.12  | 1374.22 $\pm$ 208.19     | 626.51 $\pm$ 209.13       |
> > > | Walker2d    | 4939.16 $\pm$ 107.12  | 3348.55 $\pm$ 213.20     | 3092.02$\pm$ 198.01       |
> > > | Ant         | 2815.10 $\pm$ 105.30  | 1593.23 $\pm$ 252.93     | -837.11 $\pm$ 455.39      |
> > > | CarRacing   | -87.90 $\pm$ 5.15     | -85.76 $\pm$ 2.93        | -88.01 $\pm$ 5.12         |
> > >
> > > The training curves and detailed performance analysis will be added to the revised manuscript. We would like to highlight three main findings here.
> > > - DAC shows competitive performance on Pendulum, Hopper, and Walker2D in the noise-free environment. In addition, DAC demonstrates lower sample complexity for LunarLander, Walker2d, and Ant. This is consistent with their claimed contribution on the sample complexity, where DAC leverages the off-policy RL to save the number of environment interactions.
> > >
> > > - DAC's performance drops drastically in the noise setting, especially for LunarLander and Ant. This shows that DAC, similar to GAIL and AIL, is not as robust as RHIRL in a noisy environment, which can still learn the intended behaviors from noisy expert demonstrations in the imperfect system.
> > >
> > > - DAC fails to scale up to pixel-level image input task, CarRacing. DAC cannot learn any reasonable behavior for CarRacing, where its performance falls flat to the bottom after 10k timesteps.  This shows that DAC does not fundamentally resolve the sample complexity challenge for high-dimensional, long horizon tasks.
> > >
> > > DAC improves the sample efficiency by using a replay buffer for off-policy RL training. Its replay buffer ideally should cover the entire space to better estimate the optimal policy. For mujoco tasks, the state dimension is much smaller compared to CarRacing; therefore, DAC can afford to keep pushing new transitions to the replay buffer. However, this will cause the official implementation of DAC to run into an internal memory error for CarRacing after fewer hours of training. Essentially, off-policy RL training saves sample complexity through its replay buffer. Yet, it still optimizes over the entire state space, which will cause it to run into the ``curse of dimensionality'' when the state space increases drastically. RHIRL, on the other hand, aims to break down a large global problem into sequential locally tractable problems. This allows us to scale to high-dimensional, long horizon, continuous problems previously inaccessible to the global methods.

---

> > > > ### Comment · Reviewer_VXKi · 2022-08-06
> > > > **Reviewer response**
> > > >
> > > > Thank you for running this additional experiment!
> > > >
> > > > Comparing the results with Table 3 in the Appendix, it does seem like RHIRL outperforms all baselines in the vast majority of (environment, noise level) combinations! These results go a long way in convincing me that the proposed method is better than prior methods.
> > > >
> > > > One thing that is still a bit unclear to me is _why_ the proposed method is better than these baselines. I don't entirely agree with the hypothesis that it is because RHIRL is a short horizon method, because methods like GAIL and AIRL can perform short horizon reasoning if we use a small discount factor.
> > > >
> > > > One hypothesis is that the discount factor for the GAIL and AIRL baselines is simply too large. I couldn't find the value of $\gamma$ used for these baselines in the appendix, but prior work typically uses $\gamma = 0.99$. Given that the hyperparameter search for RHIRL found that a horizon of $[15, 40]$ worked best, smaller values of $\gamma$ (perhaps between $1 - 1/15 = 0.93$ and $1-1/40 = 0.975$) might work better for these baselines.
> > > >
> > > > A second hypothesis is that the gains from RHIRL come at the cost of more resets. If, say, RHIRL resets every 20 environment steps but the baselines reset every 100 environment steps, then RHIRL would cost $5x$ more resets. Note that accounting is separate from the _cost_ of each reset, as discussed in the separate thread.
> > > >
> > > > It would be great if the authors could comment on these hypotheses, and/or offer alternative hypotheses for explaining why RHIRL outperforms the baselines.

---

> > > > > ### Author Response · Authors · 2022-08-08
> > > > > **Response to reviewer VXKi**
> > > > >
> > > > > These are two interesting hypotheses! We would like to respond to each hypothesis below.
> > > > >
> > > > > Response to Hypothesis 1:
> > > > >
> > > > > Let me first clarify the discount factor $\gamma$ used for the baselines. We use the $\gamma$ reported in the original papers for each task. Since the original papers claimed that their sets of hyperparameters are optimized for the final performance. I believe the reported $\gamma$ outperforms any smaller $\gamma$. Therefore, simply using a smaller gamma will not lead to a performance gain.
> > > > >
> > > > > One key difference between us and the baselines is that RHIRL uses the forward search for the control sequence optimization while the baselines (AIRL, GAIL) learn global value functions. Therefore, no matter how small the discounted factor is, learning a global value function causes the baselines to fail due to the "curse of dimensionality”.
> > > > >
> > > > > Response to hypothesis 2:
> > > > >
> > > > > RHIRL’s performance gain does not come from more frequent resets, but our receding horizon design. It is better than the baselines because we decompose the long horizon high-dimensional problem into a sequence of many shorter and simpler local problems. We solve each local simpler problem one at a time and stitch many local solutions to form a global trajectory. Our receding horizon algorithmic design contributes to our performance gain compared to the baselines. At the same time, the receding horizon design requires us to perform successive local optimization, which leads to more resets to the intermediate states. Therefore, more resets are not the reason for the performance gain. It is the sequence of local optimizations we perform at each reset that contributes to performance gain.

---

> > > > > > ### Comment · Reviewer_VXKi · 2022-08-09
> > > > > > **Reviewer response**
> > > > > >
> > > > > > Thanks for the clarifications about these points. I'm not sure that I entirely agree with the rebuttals to either of these hypotheses, but recognize that experimental evidence might be the only effective way to get at these questions (and it's probably not fair to expect new experiments given the short rebuttal period).

---

> > > > > > > ### Author Response · Authors · 2022-08-09
> > > > > > > **Response to reviewer VXKi**
> > > > > > >
> > > > > > > Thank you for the extended discussion on this issue! We would like to summarize theoretical and empirical evidence in support of RHIRL:
> > > > > > >
> > > > > > > - We have offered a theoretical analysis of RHIRL’s error bound as well as extensive experimental results to verify our claim on scalability and robustness.
> > > > > > >
> > > > > > > - We conducted additional experiments on the suggested DAC as an additional baseline. We outperformed DAC in the noisy environment settings and on all CarRacing tasks.
> > > > > > >
> > > > > > > We would like to highlight that the scalability of RHIRL comes from our receding horizon design, which comprises our novel objective function and a local optimal controller to optimize the proposed objective function. The experimental result on CarRacing validates our hypothesis on the difference between local vs global design. Even DAC, an off-policy IRL method optimized for better sample efficiency, fails to scale up to the pixel-level image input task. RHIRL offers a principled approach to decomposing the complex task using receding horizon design to make the previously inaccessible task solvable.

---

> > > > > > > > ### Comment · Reviewer_VXKi · 2022-08-09
> > > > > > > > **Reviewer resposne**
> > > > > > > >
> > > > > > > > Thanks for sharing these additional results, which do provide strong evidence for the good performance of the proposed method. I will take them into consideration.

---

### Official Review · Reviewer_SQCF · 2022-07-09

**Rating:** 7
**Confidence:** 3
**Soundness:** 3 good
**Presentation:** 3 good
**Contribution:** 3 good

**Summary:**

In this work, the authors propose an inverse reinforcement learning (IRL) algorithm based on trajectory optimization. As opposed to prior methods that either do single-step supervised learning (like behavioral cloning) or require reasoning about infinite horizons (like GAIL or MaxEnt IRL), the authors try to match the expert’s behavior over a short time horizon. They do this by using the MPPI algorithm with a parameterized cost function and optimize the cost function parameters to minimize the KL divergence between the control sequence from MPPI and the (unobserved) control sequences from the expert. Thanks to the structure of the cost function and KL divergence, they can efficiently optimize this objective with access only to the expert state trajectories. The authors then demonstrate that RHIRL outperforms various IRL baselines on several control tasks, including a racing task where the observation is an image.

**Questions:**

Confusion about Theorem 3.1: If K = 1 (horizon of 1 step), we should run into the same quadratic error as in behavioral cloning, but we only get linear error. Why does RHIRL not run into this issue?\
How long (in wall-clock time) does it take to run this algorithm compared to the baselines?


**Limitations:**

As implied in the “Questions”, the authors did not indicate how long in wall-clock time their algorithm takes. MPPI’s main bottleneck is the simulator, and that can make the algorithm end up running quite slowly.

**Strengths And Weaknesses:**

The proposed algorithm is a neat extension of trajectory optimization for inverse reinforcement learning, and I’m not aware of any prior method doing this.

The writing for the most part is clear. I think the authors make a nice comparison of prior IRL algorithms and where their proposed algorithm RHIRL sits among the prior work. I think the presentation in Section 3, though, is a bit spotty. For instance, the authors say they work with deterministic dynamics (line 95) but also say the dynamical system has “noisy control” (line 88).  I think it would be cleaner instead to connect to prior IRL work that assumes noisy experts, particularly those that also maximize entropy [2, 3]. In light of that, RHIRL seems to assume an optimal expert that has some Gaussian noise, which can happen under an LQG/iLQG formulation [1,4] of a maximum entropy RL problem.

One other thing I am confused about is how they make the cost function “robust” to noise. In particular, they factor the cost function into a sum of a state-dependent term and a quadratic control term. Since the noise is assumed to only enter in the controls, this structure is supposed to make the learned cost function robust to control noise, something that is corroborated in the experiments. However, do we expect this structure to generalize to other IRL problems where the noise isn’t just additive Gaussian?

The results in Section 4.4 also seem a bit weak since it is only run on the pendulum, which is a simple problem that RHIRL solves easily. There aren't any clear patterns in Fig 5, either. I would suggest running that evaluation on a more difficult problem, like Hopper or Walker2D.

Some other comments:
- The style of Fig. 1a should match that of Fig. 1b-d.
- Parentheses for exp need to be fixed in Equation (11).
- Misreferenced equations occur frequently. For example, lines 157 and 160 should reference (4).
- No anonymized code included.

[1] Todorov, E., & Jordan, M. I. (2002). Optimal feedback control as a theory of motor coordination. Nature neuroscience, 5(11), 1226-1235.\
[2] Ziebart, B. D., Maas, A. L., Bagnell, J. A., & Dey, A. K. (2008, July). Maximum entropy inverse reinforcement learning. In Aaai (Vol. 8, pp. 1433-1438).\
[3] Ho, J., & Ermon, S. (2016). Generative adversarial imitation learning. Advances in neural information processing systems, 29.\
[4] Levine, S., & Koltun, V. (2013, May). Guided policy search. In International conference on machine learning (pp. 1-9). PMLR.

---

> ### Author Response · Authors · 2022-08-02
> **Response to Reviewer SQCF**
>
> Thank you for your comments and questions! We’ll first under the questions in the Question section; then we would like to respond to the weaknesses identified by the review.
>
> Responses to the questions:
>
> Q1. Confusion about Theorem 3.1: If K = 1 (horizon of 1 step), we should run into the same quadratic error as in behavioral cloning, but we only get linear error. Why does RHIRL not run into this issue?
>
> This is a great question! The main reason is that RHIRL with K=1 is similar, but not exactly the same as behavioral cloning.  With K=1, RHIRL greedily maximizes the reward of the current step, while BC simply memorizes the optimal action for every state. This 1-step optimization gives RHIRL some ability to recover to the expert demonstrations and mitigate the compounding errors. The behavior of both algorithms on the states with expert demonstrations may be somewhat similar. However, RHIRL can handle the unseen, out-of-distribution states (near the expert demonstration) better by performing such 1-step look-ahead optimization.
>
> Q2. How long (in wall-clock time) does it take to run this algorithm compared to the baselines? MPPI’s main bottleneck is the simulator, and that can make the algorithm end up running quite slowly.
>
> This is a great point for us to clarify more! MPPI’s main computation overhead is its sampling process in the simulator. The amount of time each algorithm needs to run is nearly proportional to the number of environment interactions, as reported in figure 3 of the manuscript.
> - For Pendulum-v0 and Hopper-v2, RHIRL converges in fewer than 1/5 number of environment interactions compared to the best performing baselines.
> - For Lunarlander-v2, Walker2d, and Ant-v2, RHIRL takes comparable, if not fewer, number of environment interactions with the best performing baselines.
> - For CarRacing, RHIRL is the only method that learns a reasonable reward function, hence there are no running time comparison with other methods.
>
> Our current implementation of the sampling process is not fully parallelized on GPU. Hence RHIRL takes more wall-clock time than the baselines. However, this is not an inherent limitation of the method: a fully parallelized implementation on GPU will allow the optimal control to be computed in real-time. This is demonstrated in the MPPI paper, whose implementation can achieve control loops from 40-60 HZ using a few thousand samples of 2-3 second long trajectories, which is sufficient for real-time control.
> Next, we would like to respond to the weaknesses.
>
> Q3. Are a deterministic dynamics (line 95) and a dynamical system that has “noisy control” (line 88) contradictory to each other?
>
> There is no discrepancy. Our deterministic dynamics, as defined in equation (1), maps each (x_t, v_t) tuple to a unique next state x_{t+1}. Meanwhile, our system is corrupted by noisy control, for which the human/agent input their intended control u_t,  and the actual control fed into the deterministic dynamic is v_t \sim N(v_t|u_t, Σ). This differs from a stochastic dynamics that maps (x_t, v_t) to a probability distribution over the next state x_{t+1}.
>
> Q4. Do we expect the cost function’s structure to generalize to other IRL problems where the noise isn’t just additive Gaussian?
>
> This is an important concern! RHIRL may not generalize to a system with different types of noise. RHIRL assumes the system noises and human errors can be modeled as additive Gaussian noises. Based on this assumption, we derived a linearly separable cost function structure and learn a state cost robust to noise. Being robust to other forms of noise is beyond the scope of the current work. We agree that learning the reward function under other noises is an important extension! We would like to leave this for our future work. We would also like to highlight that, despite its simplicity, the additive Gaussian noise is fairly accurate in modeling many interesting noisy control systems for robotics [1, 2, 3, 4] and is commonly used in many optimal control literature.
>
> Q5. The results in Section 4.4 also seem a bit weak since it is only run on the pendulum, which is a simple problem that RHIRL solves easily. There aren't any clear patterns in Fig 5, either. I would suggest running that evaluation on a more difficult problem, like Hopper or Walker2D.
>
> Thank you for raising this concern!  We have observed similar trends for tasks other than pendulum-V0, though we have not systematically recorded the data. We’re currently running the ablation study on Hopper-v2, we will update the writing once it is finished.
>
> [1] H. J. Chizeck and Y. Ji, "Optimal quadratic control of jump linear systems with Gaussian noise in discrete-time,"
>
> [2] O. C. Imer and T. Basar, "Optimal control with limited controls,"
>
> [3] R. Lioutikov, A. Paraschos, J. Peters and G. Neumann, "Sample-based informationl-theoretic stochastic optimal control,"
>
> [4] O. Lev and A. Khina, "Schemes for LQG Control over Gaussian Channels with Side Information,"

---

> > ### Comment · Reviewer_SQCF · 2022-08-08
> > **Response to Response**
> >
> > Hi authors, thank you for your response! Except for what's brought up below, I'm satisfied with the response.
> >
> > Q3: From the point of view of the controller, that would make the system stochastic. It would be a particular flavor of stochasticity, in this case that the chosen control $u$ has some Gaussian noise added, which is then fed into $f$.
> >
> > I currently don't see updated results for Q5 or the code.

---

> > > ### Author Response · Authors · 2022-08-09
> > > **Response to reviewer SQCF**
> > >
> > > Thank you for the follow-up!
> > >
> > > We would like to clarify Q3 further. The additional result of the ablation study is added to appendix D.5. We apologize for the late update! We are still working on the documentation and refactoring our code base. We will release the code for the camera-ready version.
> > >
> > > >Q3.  From the point of view of the controller, that would make the system stochastic. It would be a particular flavor of stochasticity, in this case that the chosen control $u$ has some Gaussian noise added, which is then fed into $f$.
> > >
> > > Yes, definitely. Our transition has a stochastic element due to the control noise. One reason that we still want to classify our setting as a deterministic dynamic with noisy control is to distinguish it from a truly stochastic system (I’ll elaborate on later), which is a much harder problem to solve.
> > >
> > > Our system dynamic function is as follows:
> > >
> > >  $x_{t+1} = f(x_t, u_t+ \epsilon ),\epsilon \sim N(0, \Sigma)$.
> > >
> > > In our setting, the stochasticity only corrupts the control fed into the dynamic function, and it does not make the dynamic function itself stochastic. Therefore, we describe our system's dynamic function as deterministic. This is consistent with earlier work, such as MPPI, from the optimal control community.
> > >
> > > A truly stochastic system maps a state and a control to a distribution of the next state. In the presence of  control noise, such a stochastic system is of the following form:
> > >
> > > $x_{t+1} \sim P(x_{t+1}|x_t, u_t+ \epsilon ) ,\epsilon \sim N(0, \Sigma)$.
> > >
> > > This stochasticity borne to the system dynamic function brings a new challenge to the IRL methods. We have included a discussion and sketched out the extension of different IRL methods to stochastic dynamics in Appendix C.
> > >
> > >
> > > > Q4. Additional ablation study results on Hopper-v2
> > >
> > > We apologize for the late update! The additional ablation study on Hopper-v2 is added to Appendix D.5. The revised appendix has already been uploaded to reflect the changes. We have added the following text and the training curves for running RHIRL with different K on Hopper-v2 (fig 6).
> > >
> > > *"RHIRL uses the receding horizon $K$ to trade off optimality and efficiency. We hope to ablate the effect of K on Hopper-v2 to show how different $K$s affect the final performance and sample complexity. The task horizon for Hopper-v2 is 1000 steps, \ie $T= 1000$. We run RHIRL with the receding horizon $K \in \{5, 20, 100\}$. The results are consistent with the trend in the Pendulum. When $K$ is small, RHIRL improves its performance quickly but converges to the suboptimal solution. For $K = 5$, RHIRL's performance shoots up after the first few iterations to 1000, then it quickly converges to a final score of 1100. When K increases, though the performance improves slightly slower than  $K=5$, it can continue to learn and reach a score of 3071.68. At $K = 20$, it takes fewer than $1e6$ env steps to stabilize to a score greater than 3000. However, when K is too large, the learning becomes much slower. When $K= 100$, it takes more than $1e7$ env steps to stabilize to a score larger than 3000, which is 10 times more than when $K=20$. On the other hand, $K=100$ can achieve a final score of 3083, which is slightly more than that of $K=20$. The ablation study on both Hopper-v2 and Pendulum shows that our receding horizon $K$ can tradeoff optimality and efficiency: using a smaller $K$ allows us to learn faster at the expense of a sub-optimal solution, while using a large $K$ may make the learning inefficient. Seeking a suitable $K$ can balance the requirement for optimality and efficiency."*

---

> > > > ### Comment · Reviewer_SQCF · 2022-08-09
> > > > **Response**
> > > >
> > > > Hi authors,
> > > >
> > > > As a response to Q3, your transition is still stochastic, it just has a restricted form. All that is needed for a system to be stochastic is for the dynamics to have a probability distribution $p(x_{t+1} | x_t, u_t)$. This applies in our case because the system adds Gaussian noise to the control $u_t$, resulting in the stochastic dynamics $x_{t+1} = F(x_t, u_t + \varepsilon_t)$. This is in fact the perspective taken by the MPPI paper. From Section III.A. [1]:
> > > > > We consider the discrete time stochastic dynamical system $x_{t+1} = F(x_t, v_t)$. The state vector is denoted $x_t$ and $u_t$ is the commanded control input to the system. We assume that if we apply an input $u_t$ then the actual input will be $v_t \sim \mathcal{N}(u_t, \Sigma)$.
> > > >
> > > > I keep pointing this out because you don't make a distinction between the expert having Gaussian noise and the system having Gaussian noise applied to the controls. Mathematically, it is the case that they are equivalent, but for the sake of clarity it is preferable to pick one convention. Your submission keeps jumping between conventions, though:
> > > > - Line 11: "system control noise"
> > > > - Line 100: "dynamical system with noisy control"
> > > > - Line 102: "noise in expert demonstrations and in system control"
> > > > - Lines 92-93: "modeling the human error as additive Gaussian noise"
> > > >
> > > > Even though system noise and expert noise are equivalent in this case, it makes the problem setup confusing because you don't stick to one convention (i.e., either assume the expert has Gaussian noise OR the system applies Gaussian noise to the input). It gets even more confusing when Section 3.6 extends to more general stochastic dynamics, because the dynamics were already stochastic according to Section 3.1. I would recommend the following: For sections 3.1-3.5, make the assumption that the dynamical system $x_{t+1} = f(x_t, v_t)$ is deterministic and that the expert is stochastic in the sense that it samples its control from a Gaussian: $v_t \sim \mathcal{N}(u_t, \Sigma)$. I think this would make the problem setup cleaner and make the extension to stochastic dynamics less confusing, because it does become the case that you are generalizing from deterministic dynamics to stochastic dynamics.
> > > >
> > > > [1] Williams, G., Wagener, N., Goldfain, B., Drews, P., Rehg, J. M., Boots, B., & Theodorou, E. A. (2017, May). Information theoretic MPC for model-based reinforcement learning. In 2017 IEEE International Conference on Robotics and Automation (ICRA) (pp. 1714-1721). IEEE.
> > > >
> > > >
> > > > Thanks for including the extra experiment in D.5, the results are interesting and the analysis is reasonable. I think this should replace the pendulum study in the main paper as the results are much more evident.

---

> > > > > ### Author Response · Authors · 2022-08-09
> > > > > **Response to reviewer SQCF**
> > > > >
> > > > > >For sections 3.1-3.5, make the assumption that the dynamical system is deterministic and that the expert is stochastic in the sense that it samples its control from a Gaussian.
> > > > >
> > > > > That is a great suggestion! This captures our intended system and also distinguishes it with a stochastic dynamic clearly.
> > > > >
> > > > > We promise to revise our manuscript thoroughly to make this setting consistent and clear! (We apologize for not changing it now as we're afraid that the last-minute edit might introduce inconsistency accidentally)
> > > > >
> > > > > We agree that the ablation study on Hopper is more evident in showing the trade-off between optimality and efficiency. We'll add it to the revised manuscript (either by merging it with the pendulum result or by replacing the pendulum completely).

---

> > > > > > ### Comment · Reviewer_SQCF · 2022-08-09
> > > > > > **Response**
> > > > > >
> > > > > > >We promise to revise our manuscript thoroughly to make this setting consistent and clear! (We apologize for not changing it now as we're afraid that the last-minute edit might introduce inconsistency accidentally)
> > > > > >
> > > > > > That's fair. When you do make the edits for that convention, I would also recommend justifying why assuming the expert is Gaussian is reasonable, where you can make pointers to maximum entropy optimal control under LQG/iLQG assumptions [1,2,3].
> > > > > > [1] Todorov, E., & Jordan, M. I. (2002). Optimal feedback control as a theory of motor coordination. Nature neuroscience, 5(11), 1226-1235.
> > > > > > [2] Ziebart, Brian D., J. Andrew Bagnell, and Anind K. Dey. "Modeling interaction via the principle of maximum causal entropy." ICML. 2010.
> > > > > > [3] Levine, S., & Koltun, V. (2013, May). Guided policy search. In International conference on machine learning (pp. 1-9). PMLR.
> > > > > >
> > > > > > Bumping recommendation up to "Accept".

---

> > > > > > > ### Author Response · Authors · 2022-08-09
> > > > > > > **Response to Reviewer SQCF**
> > > > > > >
> > > > > > > >When you do make the edits for that convention, I would also recommend justifying why assuming the expert is Gaussian is reasonable, where you can make pointers to maximum entropy optimal control under LQG/iLQG assumptions [1,2,3].
> > > > > > >
> > > > > > > We'll definitely include this discussion to motivate our noise setting better. Thank you for pointing out these papers for our reference!  We will make sure to clean up the text for consistency and to include the suggested discussion in the final version.
> > > > > > >
> > > > > > > Thank you for all the comments and suggestions! They have helped us significantly to improve this work.

---

### Official Review · Reviewer_pMQy · 2022-07-11

**Rating:** 6
**Confidence:** 3
**Soundness:** 3 good
**Presentation:** 4 excellent
**Contribution:** 3 good

**Summary:**

This paper proposes "Receding Horizon IRL" (RHIRL), a method for IRL that is designed to (1) expose a trade off between optimality and scalability, and (2) achieve scale even in the presence of noise in the demonstrations. While many of the core subtleties of RHIRL are framed around (1)---including the use of a receding horizon---the presence of noise in the demonstrator is one of the key novelties compared to other similar approaches to IRL. In particular, the paper assumes that the demonstrator's behavior is chosen, then the observed control is actually sampled from a Gaussian with known covariance (though the paper is careful to note that knowing this covariance is not critical). Thus, the key difficult of the IRL problem posed is to recover the reward (or cost) function that the demonstrator was optimizing, even in spite of the presence of this noise in behavior. Additionally, the trajectories only include state information. To my reading of the work, the key insight comes in Section 3.3, which articulates a _local_ method for matching the parameters of the unknown reward (cost) function $\theta$ to the observed trajectory. As noted in the section, this is in contrast to many prior approaches to IRL that take a global view on this optimization. One Theorem is presented relating the KL between $p_E$ and $p_Q$ (the optimal control policy under $\theta$) to the TV between $p_E(x)$ and $p_{RHC}(x; \theta)$. Specifically, if this KL is bounded by $\epsilon$ for _every_ time step $t \in [0:T-1]$, then $D_{\text{TV}}(p_E(x) || p_{\text{RHC}}(x;\theta) < T \sqrt{\epsilon / 2}$. I will say more about this result below, but in short, the paper claims we can thus conclude that the error grows linearly with $T$. Then, Section 4 discusses experiments contrasting RHIRL to other IRL and BC methods in a variety of standard benchmarks. Results are presented in Figure 3, 5 and Tables 1 and 2---we find a clear benefit of using RHIRL across every domain in Figure 3 compared to standard approaches. Table 1 and 2 suggest this trend holds even as we scale the noise on the demonstrator's side, while Figure 5 explores the impact of modifying $K$, the size of the receding horizon (and the knob that controls the optimality vs. scalability trade off).

**Questions:**

Q1: My primary question, as stated above, is about the motivation of the setting. As described in the introduction, many other approaches to IRL focus on sub-optimal demonstrators that are _systematically_ sub-optimal, due to uncertainty about the environment or bounded compute. In contrast, this paper motivates the case where a demonstrator's actions are perturbed due to Gaussian noise. I found this latter setting harder to motivate than the former, and wondered why we should consider it rather than the more typical case of systematic sub-optimality.

Q2: In Theorem 3.1, I believe switching from KL to TV is actually making the result less clear, to me. The core of the result is translating T independent bounds over $p_E$ and $p_Q$ into a single T-step bound over $p_E$ and $p_{\text{RHC}}$. However, by moving through Pinsker's to turn the KL into TV, it makes it slightly less apparent which aspects of the resulting bound are from aggregating over the T independent bounds to form a single T-step bound, rather than which elements come from translating KL over to TV. Is there a reason to invoke TV here? Otherwise I might recommend sticking with KL all the way through (or alternatively assuming the TV is bounded by $\epsilon$ as the antecedent).

**Strengths And Weaknesses:**

[STRENGTHS]

This paper is well-written and largely well-motivated. I found the core of the method to be relatively simple to state, and the context provided in the introduction was effective at differentiating this method relative to many other approaches to IRL, imitation learning, and behavior cloning. I did find a few small aspects of the setup harder to motivate, but I will discuss that below. The paper is well organized, plots are well laid out (I found the table-like structure of Figures 2 and 3 to be a particularly effective way to convey a lot of information in a compact way). Experiments are well designed, and the results reported provide support for the conclusions drawn. For these reasons, I found this paper easy to read, and support the conclusion that RHIRL is a useful algorithm for IRL under the right assumptions.



[WEAKNESSES]

However, there were a few small aspects of the setup I had a hard time motivating. In particular, it is state early on that, unlike other approaches to IRL with a sub-optimal demonstrator, RHIRL will assume there is some Gaussian noise added to the demonstrator's behavior. This is in contrast to say, T-REX, that assumes the demonstrator is intentionally optimizing some unknown reward (cost) function, but does not know _how_, precisely, to achieve optimal behavior: "The existing works mentioned above address (i). RHIRL addresses (ii)." I appreciate the value of this distinction, and found it very useful to frame this paper. However, to me, I had a hard time motivating the setting: why would we be more interested in the case where a demonstrator adds Gaussian noise to their controls, rather than one in which the demonstrator is sub-optimal in a structurally significant way, as in the cited works that tackle "(i)"? This is my primary question about the work, which I restate below under "questions" for completeness.

Second, I found this paper to deviate heavily from standard RL notation. The notation was largely consistent (and of course is based in control, rather than RL), but as this is an IRL paper, I would find it cleaner to stick with typical RL notation. That is, to start from rewards rather than costs, and to use actions rather than controls. I suppose the connection to Gaussian noise in the demonstrator is more natural in an optimal control problem, so ultimately of course I leave this to the authors' discretion.

Lastly, one small typo: in Table 2, it looks like bold is misattributed to RHIRL in the first column of Ant.

**Overall**, I found the experimental results were well presented, the experiments were interesting, and the overall conclusions are sensible. I believe this paper will be of interest to others in the community, though I note my own skepticism about some of the motivation of the problem formulation as mentioned above. This is the primary question I would like the authors to address.

---

> ### Author Response · Authors · 2022-08-02
> **Response to reviewer pMQy**
>
> Thank you for your comments and questions! We would like to address the two questions below.
>
> Q1.  My primary question, as stated above, is about the motivation of the setting. … In contrast, this paper motivates the case where a demonstrator's actions are perturbed due to Gaussian noise. I found this latter setting harder to motivate than the former, and wondered why we should consider it rather than the more typical case of systematic sub-optimality.
>
> This is a great question! We will first discuss when our assumption is realistic and then where the limitations exist.
> We add Gaussian noises to the expert control sequences to emulate the system noise and human error. Our additive Gaussian noise setting is inspired by the rich works on optimal control [1, 2, 3, 4, 5], where the control system is often imperfect and additive Gaussian noise is sufficiently accurate to model the noisy control. Moreover, we observe that for complex control tasks, humans are good at planning the optimal path, but may fail to execute it precisely due to our limited precision over controls. Imagine that you need to use use the hand to turn the steering wheel by 5 degree, a human will not be able to achieve this exactly. We find additive Gaussian noise to be useful in modeling this type of human error.
>
> Our noise setting can be structurally more restrictive than T-REX. We would like to point out that their setting requires more training signals and human annotation effort: they have to rank the degrees of the suboptimality for all expert demonstrations in order to extrapolate the reward function, where the quality of the reward function heavily depends on the coverage of the range of the suboptimal demonstrations. In contrast, by imposing a simplified but practical structure on the form of human error, our method does not require the effort to rank the expert demonstrations.
>
> Q2. Is there a reason to invoke TV here? Otherwise I might recommend sticking with KL all the way through (or alternatively assuming the TV is bounded by ϵ as the antecedent).
>
> This is a great point for us to clarify more! TV distance is required in our final step of the derivation for its triangle inequality: it allows us to bound the distance between our global state marginal distribution and the expert by the summation of the TV distance changes over the state marginal distribution over all timesteps T. We cannot use KL for this step since KL is asymmetric and hence it does not satisfy the triangle inequality. Therefore, we use Pinsker’s inequality to upper bound the TV distance using the KL measure over the corresponding distributions and deduce the final result.
>
> [1] H. J. Chizeck and Y. Ji, "Optimal quadratic control of jump linear systems with Gaussian noise in discrete-time," Proceedings of the 27th IEEE Conference on Decision and Control, 1988, pp. 1989-1993 vol.3, doi: 10.1109/CDC.1988.194681.
>
> [2] O. C. Imer and T. Basar, "Optimal control with limited controls," 2006 American Control Conference, 2006, pp. 6 pp.-, doi: 10.1109/ACC.2006.1655371.
>
> [3] R. Lioutikov, A. Paraschos, J. Peters and G. Neumann, "Sample-based informationl-theoretic stochastic optimal control," 2014 IEEE International Conference on Robotics and Automation (ICRA), 2014, pp. 3896-3902, doi: 10.1109/ICRA.2014.6907424.
>
> [4] O. Lev and A. Khina, "Schemes for LQG Control over Gaussian Channels with Side Information," 2021 XVII International Symposium "Problems of Redundancy in Information and Control Systems" (REDUNDANCY), 2021, pp. 85-90, doi: 10.1109/REDUNDANCY52534.2021.9606456.
>
> [5] Grady Williams, Paul Drews, Brian Goldfain, James M. Rehg, and Evangelos A. Theodorou. 2018. Information-Theoretic Model Predictive Control: Theory and Applications to Autonomous Driving. Trans. Rob. 34, 6 (Dec. 2018), 1603–1622. https://doi.org/10.1109/TRO.2018.2865891

---

> > ### Comment · Reviewer_pMQy · 2022-08-05
> > **Response to author's comments**
> >
> > First, I thank the authors for their responses. To respond in more detail:
> >
> > > We add Gaussian noises to the expert control sequences to emulate the system noise and human error.
> >
> > Personally, I suspect the kinds of errors people make in sequential decision making likely tend not to Gaussian, but rather admit other structural regularities related to planning, resource constraints, uncertainty, or the sorts of resource-rationality and boundedly rational pressures. I am thinking broadly along the lines of Kahneman and Tversky's "Prospect theory: An analysis of decision under risk."
> >
> > However, that being said, Gaussian noise is still potentially a natural choice to make, and as we do not yet have idealized models of the sources of human error in sequential decision making, it is useful to study this setting. I found comments of the author's relating this model to well known settings in optimal control to be convincing, too. I do believe the paper would be improved with more upfront discussion of this assumption.
> >
> > >  We cannot use KL for this step since KL is asymmetric and hence it does not satisfy the triangle inequality.
> >
> > Right, for that reason I believe a cleaner version of the result might assume a bounded TV as the antecedent rather than go through Pinsker's. In other words, why start by assuming the KL is bounded?
> >
> > I will continue to read the other reviews and replies and think about the overall evaluation, but at present I am inclined to retain a positive recommendation of the paper.

---

> > > ### Author Response · Authors · 2022-08-08
> > > **Response to reviewer pMQy (Part 2)**
> > >
> > > We agree that using one kind of distance measure throughout the proof will make the theoretical derivation cleaner. As the triangle inequality is required for the proof, we will need to minimize the TV distance over the local control sequence distribution. One reason that we may still prefer to minimize the KL divergence over the local control sequence space is that KL divergence matches with the objective function of MPPI, a competitive, sampling-based optimal controller that scales to high-dimensional continuous tasks. Therefore, using the KL divergence for local control optimization leads to strong optimization results. To make the transition from KL to TV more natural, we have revised the manuscript (line 236 -240) and added the motivation for such transition. We thank the reviewer for highlighting this issue! The revision in Section 3.5 is as follows:
> > >
> > > *"Note that our local optimization's objective is defined in KL divergence, while the final error bound is in TV distance. We switch the distance measures to get the best of both: minimizing the KL divergence leads to strong local optimization results but KL itself is not a proper metric. Therefore, we further bound the KL divergence by TV distance to obtain a proper distance bound for the final result."*

---

> > > ### Author Response · Authors · 2022-08-08
> > > **Response to reviewer pMQy (part 1)**
> > >
> > > We completely agree that the types of errors humans commit for sequential decision-making can be much richer and more complex than Gaussian additive noise. Thank you for mentioning Kahneman and Tversky's "Prospect theory: An analysis of decision under risk." This is indeed very interesting for our future work.
> > >
> > > We do believe that the human errors in sequential decision-making exhibit certain structural regularity related to planning under uncertainty and bounded rationality … Studying and exploiting these structures allows us to understand the intentions of human behaviors better. We have revised the Related Work (line 90 - 97) to state our current noise setting clearly, and to better motivate the discussion for future work on more specialized human error modeling. For example, humans are prone to make decisions myopically in sequential decision-making tasks due to bounded rationality. It will be interesting to learn the globally optimal cost function (maybe a hierarchical one) from locally optimal expert demonstrations. The revision of the Related Work (line 90 - 97) is as follows:
> > >
> > > *"Modelling the human error as additive Gaussian noise is a natural choice technically, but compromises on realism. Human errors in sequential decision-making may admit other structural regularities, as a result of planning, resource constraints, uncertainty, and bounded rational pressures [1]. Studying the specific forms of human errors in the context of IRL requires insights beyond the scope of our current work, and it is a promising direction for future IRL works."*
> > >
> > > [1] Kahneman, D., & Tversky, A. (1979). Prospect Theory: An Analysis of Decision under Risk. Econometrica, 47(2), 263–291. https://doi.org/10.2307/1914185

---

### Official Review · Reviewer_xWf5 · 2022-07-11

**Rating:** 6
**Confidence:** 4
**Soundness:** 2 fair
**Presentation:** 2 fair
**Contribution:** 3 good

**Summary:**

This paper proposes a receding horizon IRL algorithm to address scalability for reward learning in high-dimensional control tasks. By using model-based planning and optimizing the learned reward function to match demonstrations locally, the authors show that their approach can perform comparably or better than prior works.

**Questions:**

Missing very related work
MacGlashan and Littman. "Between imitation and intention learning." Twenty-Fourth International Joint Conference on Artificial Intelligence. 2015.
They are, to my knowledge, the first to formulate a receding horizon IRL algorithm. The authors should add this reference and adjust their claims accordingly.

Authors claim that prior work isn't robust to noise, but MaxEnt IRL and Bayesian IRL are both robust to noisy demos.

Other related work is

Levine, Sergey, and Vladlen Koltun. "Continuous inverse optimal control with locally optimal examples." arXiv preprint arXiv:1206.4617 (2012).

and

Kalakrishnan, Mrinal, et al. "Learning objective functions for manipulation." 2013 IEEE International Conference on Robotics and Automation. IEEE, 2013.

which both consider local optimization to make IRL more tractable.

Guided cost learning also performs local optimization over many steps. How is the proposed approach different?

Line 251: Authors claim that no one else has looked at visual control tasks for IRL, but there has been prior work in this area including [12] and [27] (cited in the submission) as well as

Uchibe, Eiji. "Model-free deep inverse reinforcement learning by logistic regression." Neural Processing Letters 47.3 (2018): 891-905.

Section 4.4 How is task horizon chosen for the other tasks not considered in this ablation? How does it compare with the full horizon?


If you do finite horizon how do you deal with the cost to go at the last state? This is often learned for MPC style approaches.

Also, the learned cost function may be very different from the true cost.
Jain, Avik, et al. "Optimal Cost Design for Model Predictive Control." Learning for Dynamics and Control. PMLR, 2021.
showed that learning shaped rewards for MPC worked better than using the true rewards.  Does this cause problems when the rewards are learned from data? I.e. is the reward significantly affected by the limitations of MPC and not suitable for optimization with other methods?
Adding discussion about this would be good.

When K=1 does RHIRL reduce to BC?

Many of the figures are too small and make reading difficult.

**Limitations:**

Yes.

**Strengths And Weaknesses:**

+ Strong empirical results on a variety of domains

+ Good results on robustness to noise in demonstrations


- Relevant prior work is missing or not discussed in enough detail.

- I did not see any results on the computational time required for RHIRL vs other methods compared to

---

> ### Author Response · Authors · 2022-08-02
> **Response to reviewer xWf5**
>
> Thank you for your comments and questions!
>
> Q1. Discussion on the relevant prior work:
>
> 1/ "Between imitation and intention learning." [1]
> We thank the reviewer for pointing out this important reference. We carefully checked several recent surveys (Osa et al., 2018; Arora and Doshi 2021), but unfortunately, still missed this one.
>
> [1] and ours share a similar high-level idea of using MPC as the optimizer to reduce the computation complexity of reward learning. We have moderated our claim and acknowledged the earlier idea in the revised manuscript.
>
> However, our IRL method and [1] differ in the key technical and algorithmic choices that lead to the scalability and robustness of our method:
> - A structured representation of the task cost, and
> - MPPI as the optimizer, over a chosen finite horizon
>
> which are required to work TOGETHER, to enable our method to handle high-dimensional, long-horizon tasks and noisy control input. In comparison, [1] parameterizes its reward function as a linearly weighted state feature vector. It chooses the state features manually. Our method models the reward function as a neural network that takes the raw observation as the input and bypasses the manual state feature selection. Moreover, our work focuses on scaling IRL to high-dimensional, long-horizon tasks, through the use of MPPI, a sampling-based MPC controller. [1] would likely run into difficulty with high-dimensional systems as it uses a recursive tree-search algorithm to solve the Q-value gradient.
>
> 2/ MaxEnt IRL and Bayesian IRL are robust to noisy demos. How are the noise settings differ?
>
> MaxEntIRL and BIRL assume that the probability of a sub-optimal demonstration decreases exponentially with the level of sub-optimality. Roughly, most of the demos are near-optimal. This differs from our noise setting, where the control input is corrupted at every step. Under the assumption of independent additive noise, near-optimal demonstrations occur with small probability.
>
> 3/ How are local optimization methods [2, 3] differ from RHIRL? What about GCL?
>
> [2] Continuous inverse optimal control with locally optimal examples\
> [3] Learning objective functions for manipulation
>
> RHIRL and the [2] learn a local reward function. However, [2] learns a local reward function that only induces a locally optimal trajectory and has no guarantee over the global optimality of the trajectory. In contrast, RHIRL aims to induce a globally optimal trajectory by optimizing the local reward function to match the globally optimal expert through replanning.
>
> Though [3] and GCL optimize the trajectory using local optimization, they update the reward function over the global trajectory distribution. In contrast, RHIRL performs both control optimization and the reward update over the local control sequence space. Two methods contrast sharply in their scalability on long-horizon tasks. Optimizing over the long sequence trajectory space is time-consuming and susceptible to high variance while using a shorter local control sequence makes the reward update tractable.
>
> Q2. Section 4.4 How is K chosen for the other tasks not considered in this ablation?
>
> They are chosen by grid search. The optimal receding horizons are always shorter than the entire task horizon. Especially for long horizon mujoco tasks where T=1000, the optimal receding horizons range from 15-30, which are small fractions of the full task horizons.
>
> Q3. How do you deal with the cost to go at the last state?
>
> The cost of the last state is learned just like any other state. We did not treat the end state differently since the end state for this iteration may be the intermediate state of future iterations.
>
> Q4.  Do you learn a shaped reward function?  What implications?
>
> This is a great question! RHIRL learns a shaped reward function. This allows RHIRL to induce trajectories similar to a globally optimal expert, while MPC using the true reward function may be stuck at the local optimal. Our reward function is learned locally. This implies that this local reward function may be incompatible with the global optimizer (RL). This is the trade-off between optimality and scalability. For high-dim long horizon tasks, where optimizing policy directly can be intractable, RHIRL can learn a shaped local reward function that allows efficient short horizon planning to induce a global optimal trajectory.
>
> Q5. When K=1, does RHIRL reduce to BC?
>
> No. With K=1, RHIRL greedily maximizes the reward of the current step, while BC simply memorizes the optimal action for every state. This 1-step optimization gives RHIRL some ability to recover back to the expert demonstrations and mitigate the compounding errors. The behavior of both algorithms on the states with expert demonstrations may be somewhat similar. However, RHIRL can handle the unseen, out-of-distribution states (near the expert demonstration) better by performing such 1-step look-ahead optimization.
>
> Q6. We will adjust the figures in the manuscript.

---

> > ### Comment · Reviewer_xWf5 · 2022-08-09
> > **response**
> >
> > Thank you for your clarification and response. I think the changes to the paper and clarifications have improved the submission and I have adjusted my score to a weak accept.

---

> > > ### Author Response · Authors · 2022-08-09
> > > **Response to Reviewer  xWf5**
> > >
> > > Thank you for pointing out that important reference, as well as offering several other comments and suggestions! They have helped us significantly in improving this work.

---

### Author Response · Authors · 2022-08-02
**Response for all reviewers**

We thank all reviewers for the encouragement and the suggestions for improvement:
- A simple and effective IRL method
- Strong empirical results on a variety of domains
- Good results on robustness to noise in demonstrations

We thank Reviewer xWf5 for pointing out an important reference. We added the reference to the revised manuscript and acknowledged the earlier idea. While our IRL method and the earlier work share one high-level idea, they differ in the key technical and algorithmic choices that lead to the scalability and robustness of our method. See the detailed response below.

We’ve revised the manuscript according to the reviewers’ comments, including, particularly, the additional references.

---

### Author Response · Authors · 2022-08-09
**Response to all reviewers**

We thank all reviewers for reviewing and recognizing the strengths of this work!

- Novelty: “The proposed algorithm is a neat extension of trajectory optimization for inverse reinforcement learning, and I’m not aware of any prior method doing this.” (reviewer  SQCF)

- Simplicity and clarity: “... the core of the method to be relatively simple to state” (reviewer pMQy)

- Experimental results: “Strong empirical results on a variety of domains” (reviewer xWf5)

- Presentation: “This paper is well-written and largely well-motivated.” (reviewer pMQy)

- “Overall, I think the paper is generally strong. The proposed method seems theoretically sound, and empirically seems to outperform some baselines.” (reviewer VXKi)

We received many great suggestions and revised the manuscript accordingly. Thank you very much to you all for helping us improve this work!

We believe we adequately addressed the most critical review (VXKi), particularly, the 3 questions highlighted in the summary section:

>“I'm not clear about the difference between the proposed method and MaxEnt IRL + MPPI

We clarified this issue during the discussion period.

>I'm not sure whether the proposed method requires resetting the environment, in ways that the alternative approaches don't

We revised the text (Sect 3.1) to explicitly state resetting requirements, in comparison with earlier work.

> It's not clear how the proposed method would compare to more recent/competitive imitation learning methods.

 We conducted experiments on DAC, which the reviewer specifically asked for. The results show RHIRL outperforms DAC, as expected. While DAC improves the sample efficiency by using a replay buffer for off-policy RL training, it still optimizes over the entire state space and runs into the ``curse of dimensionality'' when the state space dimensions increase drastically. RHIRL optimizes locally in a receding horizon manner and “stitches” together the local solutions to learn a cost function.  It thereby avoids the “curse of dimensionality”. This is one key idea behind RHIRL.

According to the reviewer SQCF’s comment, we added an additional ablation study on the more complex task, Hopper. The results are consistent with those earlier on Pendulum.

Based on Reviewer pMQy’s comments, we clarified our assumption on the nature of noise in the system under study (Sect. 2) and the choice of error metric in the theoretical analysis (Sect. 3.5).

We thank Reviewer xWf5 for pointing out an important reference. We added the reference to the revised manuscript and acknowledged the earlier idea. While our IRL method and the earlier work share one high-level idea, they differ in the key technical and algorithmic choices that lead to the scalability and robustness of our method. See the detailed response below.

---

> ### Comment · Reviewer_VXKi · 2022-08-09
> **Thanks for the summary**
>
> Thanks for posting this summary! I haven't seen this done before, but it's honestly really helpful as a reviewer to see a summary of the entire discussion.
>
> FYI: The top/bottom margins on the current paper seem too big.

---

> > ### Author Response · Authors · 2022-08-09
> > **Thank you for taking time to review our paper**
> >
> > You are welcome!
> >
> > Thank you for taking the time to review and help us improve this work.
> >
> > > FYI: The top/bottom margins on the current paper seem too big.
> >
> > Thank you for the reminder! We will adjust the format for the final version.

---

### Meta-Review · Area_Chair_xuqY · 2022-08-25

**Recommendation:** Accept
**Confidence:** Less certain

**Metareview:**

This paper presents a new algorithm for inverse reinforcement learning (IRL) that uses receding horizon to locally match expert demonstrations, which generally results in better performance compared to global methods, especially with long horizons. The algorithm is shown to outperform some state of the art alternatives on a few simulated robotic environments.
The paper is overall well-written, technically strong, and the proposed idea is interesting. The main issue, raised by one of the reviewers, is the assumption that the algorithm has access to a resettable dynamics model, but it seems like this assumption is also made by other model-free methods that are compared here, such as GAIL.

**Award:**

No

---

### Decision · Program_Chairs · 2022-09-14

Accept